*Report*

# Tomatidine is a senotherapeutic compound that improves cognitive function and reduces cellular senescence in aged mice

Daniela G Costa[1,2,3,4,5,12], Lucy M Gee[1,2,12], Gung Lee[1,2], Lilian Sales Gomez [1,2], Ana Catarina Franco [1,2], Maria Grazia Vizioli[1,2], Karla Valdivieso [1,2], Lei J Zhang [6,7], Nick Pirius[1,2], Helene Martini [1,2], Rebecca A Porritt[8], Dominik Saul [9], Claudia Cavadas [3,4,5], Scott Ebert [10], Nathan K LeBrasseur [1,2], Sundeep Khosla [1,2,10], João F Passos [1,2], Christopher M Adams[10], Paul D Robbins [7] & Diana Jurk [1,2,11]✉

## Abstract

**Cellular senescence drives aging and age-related dysfunction across multiple tissues, including the brain. Through a high-content, senescent cell-based phenotypic screen of a small panel of natural products, we identified tomatidine, an aglycone of tomatine found in tomatoes, as a previously unrecognized senotherapeutic agent. In senescent human brain microvascular endothelial cells and fibroblasts, tomatidine selectively suppressed SASP expression without affecting p16[Ink4a] or p21[Cip1] levels consistent with a senomorphic effect. In aged mice, tomatidine reduced frailty and improved motor coordination and cognitive performance. These functional benefits were accompanied by reduced senescence markers (p16 [Ink4a], p21 [Cip1], and telomere-associated DNA damage foci) in liver, skin, and hippocampal neurons, along with decreased neuroinflammation and microglial activation. Tomatidine also diminished brain endothelial cell senescence while enhancing tight junction protein expression, suggesting preserved blood–brain barrier integrity. Together, these findings identify tomatidine as a promising senescence-targeting compound with beneficial effects in aged mice and support its further evaluation in mechanistic and translational studies.**

**Keywords** Brain; Aging; Senescence; Senomorphic; Cognition
**Subject Categories** Autophagy & Cell Death; Neuroscience; Pharmacology & Drug Discovery

## Introduction

Cellular senescence is a state of stable proliferative arrest that occurs in response to various forms of cellular stress, including DNA damage, oncogene activation, and telomere dysfunction (Gorgoulis et al, 2019). Although initially beneficial, limiting tumorigenesis and stimulating tissue repair, senescent cells accumulate with age and can disrupt tissue homeostasis partially through the secretion of pro-inflammatory factors collectively known as the senescence-associated secretory phenotype (SASP) (Campisi and d'Adda di Fagagna, 2007; Coppé et al, 2010). A growing body of evidence suggests that this accumulation contributes to multiple age-related pathologies, and targeting senescent cells has emerged as a promising strategy to improve healthspan and delay the onset of age-associated diseases (Robbins et al, 2021).

In the central nervous system, senescent-like phenotypes have been observed in various cell types, including neurons, glial cells, and brain endothelial cells (Jurk et al, 2012; Melo Dos Santos et al, 2024). Senescence in the brain has been linked to neuroinflammation, blood–brain barrier dysfunction, and impaired neurogenesis, processes implicated in age-related cognitive decline and neurodegenerative disorders (Baker and Petersen, 2018; Novo et al, 2024; Ogrodnik et al, 2021; Ogrodnik et al, 2019). Our own work has shown that targeting senescent cells through either genetic or pharmacological approaches improves age-related cognitive function (Ogrodnik et al, 2021), as well as reducing blood–brain barrier permeability (Novo et al, 2024). As a result, targeting senescence may hold therapeutic potential for maintaining cognitive function during aging.

[1]Department of Physiology and Biomedical Engineering, Mayo Clinic, Rochester, MN 55905, USA. [2]Robert and Arlene Kogod Center on Aging, Mayo Clinic, 200 First Street SW, Rochester, MN 55905, USA. [3]Center for Neuroscience and Cell Biology (CNC-UC), University of Coimbra, Coimbra, Portugal. [4]Faculty of Pharmacy, University of Coimbra, Coimbra, Portugal. [5]Center for Innovation in Biomedicine and Biotechnology (CIBB), University of Coimbra, Coimbra, Portugal. [6]Department of Pharmaceutical and Biomedical Sciences, College of Pharmacy, University of Georgia, Athens, GA, USA. [7]Masonic Institute on the Biology of Aging and Metabolism, Department of Biochemistry, Molecular Biology and Biophysics, University of Minnesota, Minneapolis, MN, USA. [8]Sanford Burnham Prebys Medical Discovery Institute, La Jolla, CA, USA. [9]Robert Bosch Center for Tumor Diseases, Stuttgart, Germany. [10]Division of Endocrinology, Diabetes, Metabolism and Nutrition, Mayo Clinic, Rochester, MN, USA. [11]Department of Neurology, Manyo Clinic, Rochester, MN 55905, USA. [12]These authors contributed equally: Daniela G Costa, Lucy M Gee. ✉E-mail: jurk.diana@mayo.edu

Efforts to develop senescence-targeting therapies have led to the identification of two major classes of compounds: senolytics, which selectively eliminate senescent cells, and senomorphics, which attenuate features of senescence such as the SASP without inducing cell death (Zhang et al, 2023). High-throughput screening approaches and preclinical testing in aged animals have uncovered a growing list of candidate senotherapeutics (Zhang et al, 2021; Zhang et al, 2024; Zhang et al, 2026). However, the number of compounds with demonstrated in vivo efficacy, especially in brain aging, remains limited.

In this study, we identify tomatidine, a natural steroidal alkaloid, as a novel senotherapeutic agent and characterize its therapeutic effects across multiple tissues, with a focus on cognitive function and brain aging.

# Results

## Identification of tomatidine as a senotherapeutic compound

One emerging strategy to counteract aging is the development of senotherapeutic compounds, either senolytics, which selectively eliminate senescent cells, or senomorphics, which attenuate key features of the senescent phenotype, such as SASP expression or senescence-associated β-galactosidase (SA-β-gal) activity, while preserving cell viability (Zhang et al, 2023). To this end, we conducted a high-content, senescent cell-based phenotypic screen of a small panel of natural products with reported health benefits on senescent Ercc1-deficient MEFs using SA-beta-gal as a primary endpoint detected by $C_{12}FDG$ (Fuhrmann-Stroissnigg et al, 2017; Zhang et al, 2026). Primary $Ercc1^{-/-}$ mouse embryonic fibroblasts (MEFs) were employed as a senescent cell model. Due to their deficiency in ERCC1, a component of the ERCC1-XPF endonuclease complex required for multiple DNA repair pathways, these MEFs are highly prone to senescence under oxidative stress. This makes them a robust model for studying senescence-associated responses, including stress induced by serial passage at 20% $O_2$. Wild-type (WT) MEFs, which retain intact proliferative and DNA repair capacity, served as non-senescent controls. Cells were treated with compounds for 48 h, followed by staining with the fluorogenic substrate $C_{12}FDG$ to measure SA-beta-gal activity. Senescence levels were quantified as the percentage of $C_{12}FDG$-positive cells using a high-content fluorescence imager (Fig. EV1A,D). To discriminate between senomorphic and senolytic effects, we determined the $EC_{50}$ values for both the fraction of $C_{12}FDG$-positive $Ercc1^{-/-}$ MEFs and for the total cell number. Concentrations that reduced both senescent cell burden and overall cell numbers were classified as senolytic, while those that selectively reduced $C_{12}FDG$ positivity without affecting total cell count were categorized as senomorphic. Selectivity index (SI) was calculated as the ratio of $EC_{50}$ values in non-senescent versus senescent cells, providing a measure of compound selectivity.

From the small screen, tomatidine, an aglycone of tomatine naturally found in tomatoes, was identified as a promising candidate with both senolytic and senomorphic activity in a concentration-dependent manner (Fig. EV1A–D). Tomatidine reduced the percent of $C_{12}FDG$-positive cells (senescent $Ercc1^{-/-}$

MEFs) with an $EC_{50}$ value of 1.241 μM. At low concentrations, there was no reduction in the cell number of $Ercc1^{-/-}$ MEFs, consistent with a senomorphic effect, whereas at high doses there was a reduction in the total number of cells, consistent with a senolytic effect ($EC_{50} = 16.73$ μM). Tomatidine had only a marginal effect on non-senescent WT MEFs at lower concentrations ($EC_{50} = 41.52$ μM), yielding a good selectivity index (SI) of 33.5 (Fig. EV1C). Given the known safety of tomatidine as a naturally occurring compound (Dyle et al, 2014), its oral bioavailability, and prior evidence linking it to improved mitochondrial function, inhibited skeletal muscle atrophy and reduced inflammation in aging models, we further examined the senotherapeutic effects of tomatidine (Ahsan et al, 2020; Ebert et al, 2015; Fang et al, 2017).

We next tested tomatidine in two distinct senescent human cell types: lung fibroblasts (IMR90) and human brain microvascular endothelial cells (HBMEC), both induced into senescence by X-ray irradiation. In neither cell type did tomatidine exhibit senolytic activity; however, it significantly reduced the expression of SASP-related genes without altering p21$^{CIP1}$ or p16$^{INK4a}$ levels, consistent with a senomorphic effect (Fig. EV1E–I). Treatment of senescent HBMECs showed no significant reduction in p21$^{CIP1}$ and p16$^{INK4a}$ when treated with tomatidine at day 10 for 48 h (Fig. EV1J,K), again confirming that tomatidine has no senolytic activity on certain senescent cell types.

## Tomatidine improves healthspan and cognitive function in aged mice

To assess the impact of tomatidine on organismal aging, we supplemented the diet of aged mice (21 months old) with 0.05% tomatidine for 3 months (Fig. 1A). Tomatidine-treated mice showed a significant reduction in frailty index (Whitehead et al, 2013), a composite score that integrates age-associated changes across multiple domains, including coat condition, gait, reflexes, and body composition (Fig. 1B). Importantly, throughout the 3-month intervention, tomatidine supplementation did not lead to any detectable adverse or negative side effects, as treated mice maintained normal behavior, body weight trajectory, and overall health.

Neuromuscular coordination, assessed by the pole test, a measure of motor function and balance, was significantly improved (Figs. 1C and EV2A). In contrast, performance on the Rotarod test, which measures the time mice remain on a rotating rod, showed no significant difference between groups (Fig. EV2B,C).

Cognitive performance was first evaluated using the Y maze, where increased time spent in the novel arm reflects enhanced spatial working memory. Tomatidine-treated mice demonstrated significant improvements in this task (Fig. 1D,E). Similarly, in the Stone T maze, a test of spatial learning and memory, tomatidine-treated mice made significantly fewer errors, reaching performance levels comparable to young controls (Figs. 1F,G and EV2D,G–J). While the overall time required to complete the maze did not differ between groups (Fig. EV2E), treated mice displayed enhanced learning across trials. Specifically, they showed significant reductions in errors between trials 1–3 vs. 7–9 and 4–6 vs. 7–9, whereas untreated mice improved only between trials 1–3 vs. 7–9 (Fig. EV2D).

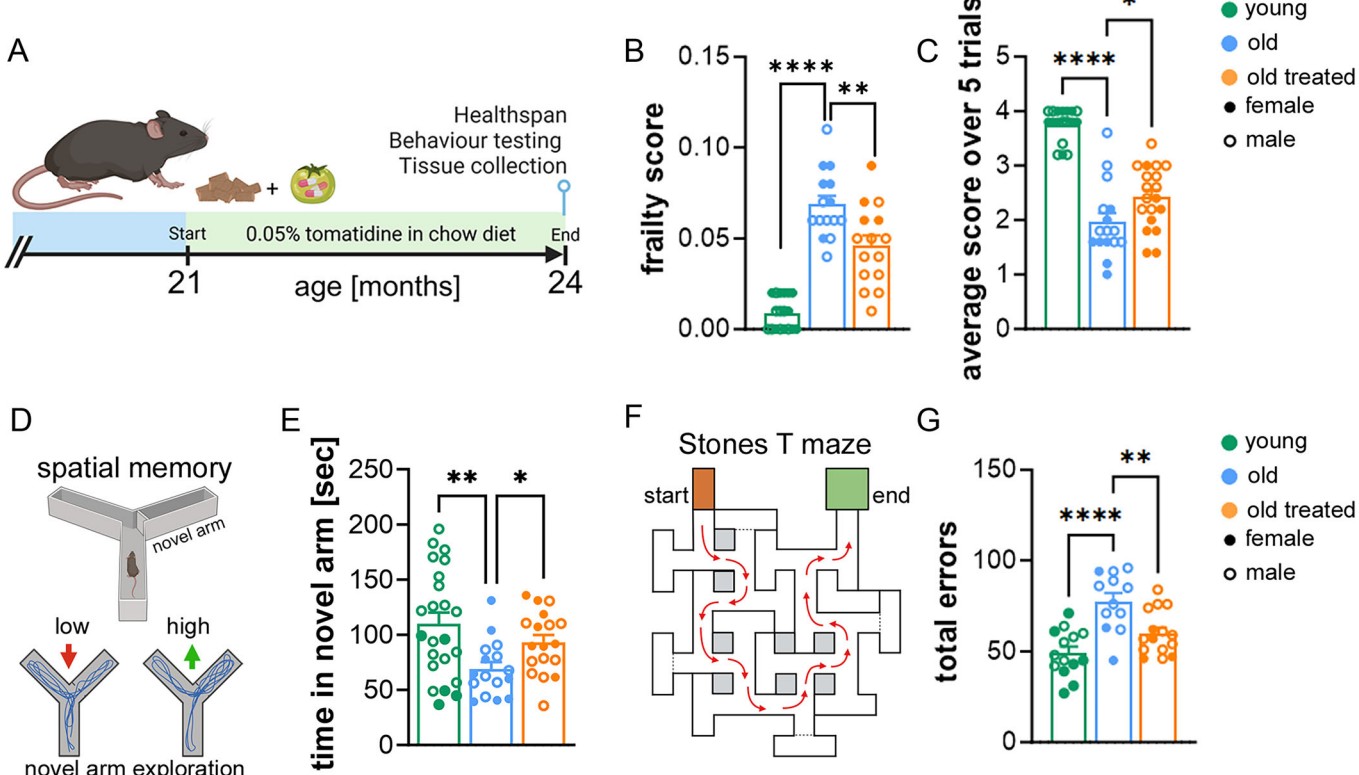

**Figure 1. Behavioral and physiological assessments in aged mice treated with tomatidine.**

(A) Schematic of the experimental design and behavioral testing timeline. (B) Frailty scores assessed using a 27-item frailty index in young control mice, untreated aged mice, and aged mice treated with tomatidine ($p < 0.0001$ and $p = 0.0018$). (C) Performance on the pole test, shown as time spent on the pole, in young, untreated aged, and tomatidine-treated aged mice ($p < 0.0001$ and $p = 0.0307$). (D) Schematic of the Y maze setup; spatial working memory was evaluated by measuring time spent in the novel arm. (E) Time spent in the novel arm during the Y maze test in young, untreated aged, and tomatidine-treated aged mice ($p = 0.0048$ and $p = 0.0354$). (F) Schematic of the Stone T maze indicating the paths and error points used for assessment of learning and memory. (G) Number of errors made during the Stone T maze testing in young, untreated aged, and tomatidine-treated aged mice ($p < 0.0001$ and $p = 0.0049$). Data were presented as mean ± s.e.m. Statistical significance was determined using one-way ANOVA followed by Tukey's multiple comparisons test. $n = 12$–23, biological replicates. Males are represented by open circles, females by filled circles. Source data are available online for this figure.

These findings identify tomatidine as a promising senotherapeutic compound with both in vitro and in vivo efficacy. Following these observations, we sought to examine how it impacts senescence at the tissue level.

## Tomatidine reduces senescence markers in the liver and skin

To gain mechanistic insight into the effects of tomatidine in vivo, we examined established markers of cellular senescence, including the cyclin-dependent kinase inhibitors p16[Ink4a] and p21[Cip1], which are key regulators of senescence-associated growth arrest, and telomere-associated DNA damage foci (TAF), which mark persistent DNA damage at telomeres and are a hallmark of senescent cells in aging tissues (Hewitt et al, 2012). In the liver, tomatidine treatment led to a significant reduction in p16[Ink4a] and p21[Cip1] mRNA expression in hepatocytes, as assessed by RNA in situ hybridization (RNA-ISH), and a marked decrease in TAF, measured by the colocalization of γH2A.X and telomeres (Fig. 2A–E). Similar results were observed in the skin, where tomatidine significantly reduced p16[Ink4a], p21[Cip1], and TAF levels in

both the epidermis and dermis (Fig. 2F–J), suggesting a broad reduction in senescent cell burden across multiple tissues. However, tomatidine treatment did not significantly alter skin thickness, a parameter that normally declines with age (Fig. EV3A–C).

## Tomatidine attenuates neuronal senescence and neuroinflammation in the aged brain

Given the cognitive improvements observed in tomatidine-treated aged mice compared to controls, we next investigated its effects on brain senescence and inflammation. We focused on the hippocampus, a region critical for spatial and working memory, and assessed cellular senescence by analyzing the expression of p16[Ink4a] and p21[Cip1], and the number of TAF in neurons of the CA3 region. CA3 plays a unique role in memory processing, and its neurons exhibit both structural and functional decline with aging (Fielder et al, 2020). Tomatidine treatment significantly reduced the frequency of neurons positive for each of these senescence-associated markers (Fig. 3A–C). Importantly, this reduction in neuronal senescence did not reflect loss of neurons,

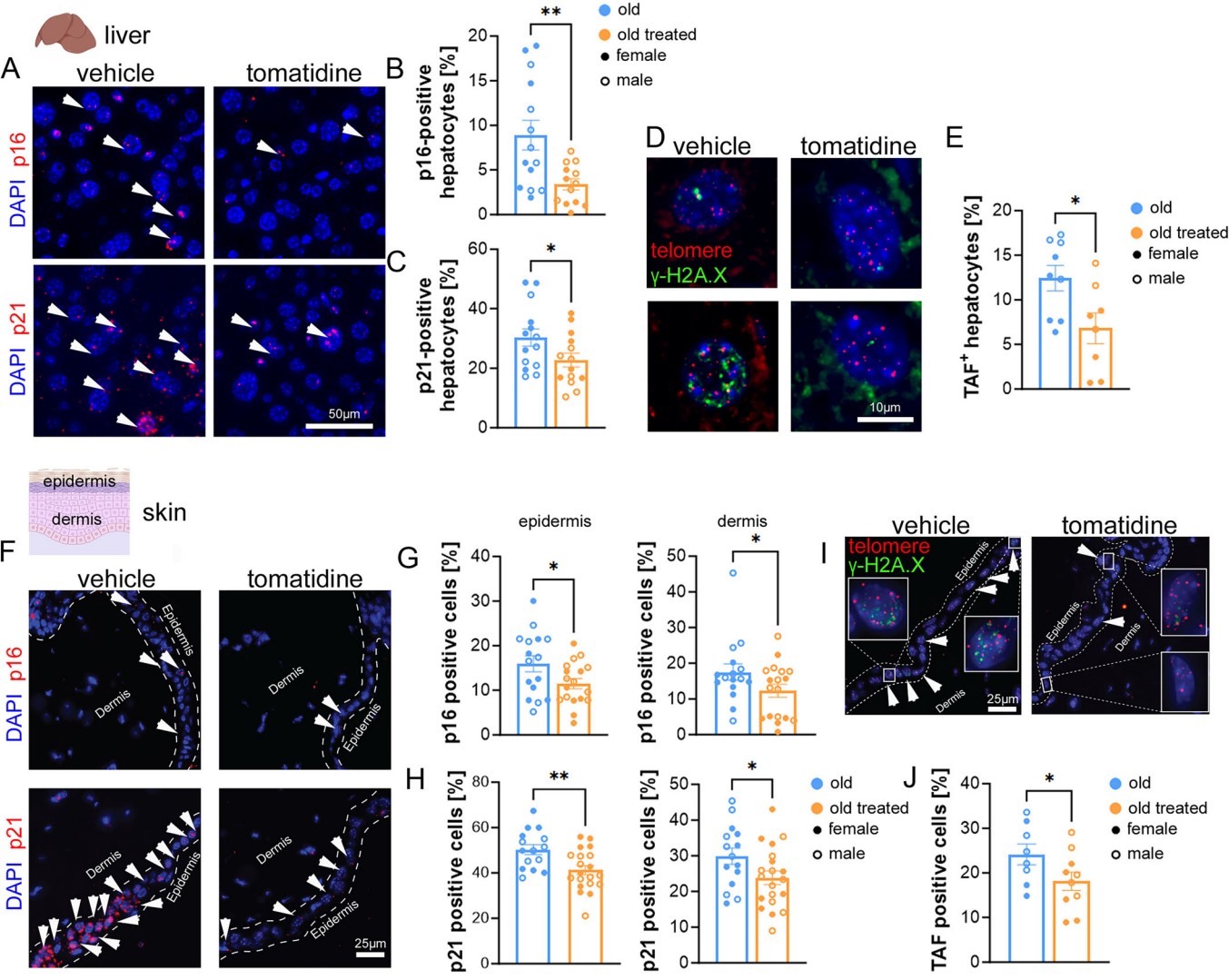

**Figure 2. Senescence markers in liver and skin from aged mice treated with tomatidine.**

(A) Representative RNA-ISH images showing p16 and p21 expression in paraffin-embedded liver sections (blue: DAPI; red: p16 or p21), white arrows indicate positive cells. (B, C) Quantification of p16-positive ($p = 0.0028$) and p21-positive hepatocytes ($p = 0.0272$), respectively. (D) Representative images of telomere-associated DNA damage foci (TAF) in liver sections (blue: DAPI; red: telomeres; green: γ-H2A.X). (E) Quantification of TAF per hepatocyte ($p = 0.0113$). (F) Representative RNA-ISH images showing p16 and p21 expression in skin sections (blue: DAPI; red: p16 or p21), white arrows indicate positive cells. (G, H) Quantification of p16-positive and p21-positive cells in epidermis and dermis. $p$ values for comparisons between groups in the epidermis (p16: $p = 0.0387$; p21: $p = 0.0022$) and dermis (p16: $p = 0.0415$; p21: $p = 0.0251$) are indicated. (I) Representative TAF staining images in skin (blue: DAPI; red: telomeres; green: γ-H2A.X). Magnified views of framed regions are shown; white arrows indicate TAF-positive cells. (J) Quantification of TAF per epidermal cell ($p = 0.0369$). Data were presented as mean ± s.e.m. Statistical significance was determined using an unpaired $t$-test. $n = 8–21$, biological replicates. Males are represented by open circles, females by filled circles. Scale bars (A) 50 µm, (D) 10 µm, and (F, I) 25 µm. Source data are available online for this figure.

as neuronal density in both the dentate gyrus and CA1 region remained unchanged (Fig. EV4A,B). Notably, the decrease in senescence markers was not restricted to the hippocampus, as similar reductions were also observed in the cortex (Fig. EV4E,F).

To assess neuroinflammation, we analyzed Iba1$^+$ microglia. In aged mice, tomatidine reduced both microglial number and soma size in the hippocampus (Fig. 3D,E), consistent with attenuated microglial activation. Comparisons with young mice showed the expected age-related increase in microglial number and soma size in the hippocampus, and an age-related increase in microglial

number, but not soma size, in the cortex (Fig. EV4C–F). Tomatidine partially reversed these age-associated changes. In addition, cytokine profiling of hippocampal homogenates revealed broad reductions in pro-inflammatory factors in tomatidine-treated mice (Fig. 3F,G). Together, these findings indicate that tomatidine alleviates neuronal senescence and neuroinflammation in the aged brain.

Given these improvements in neuronal senescence and inflammation, we next asked whether tomatidine also impacts brain endothelial cells (BECs), which form the structural and functional basis of the blood–brain barrier (BBB).

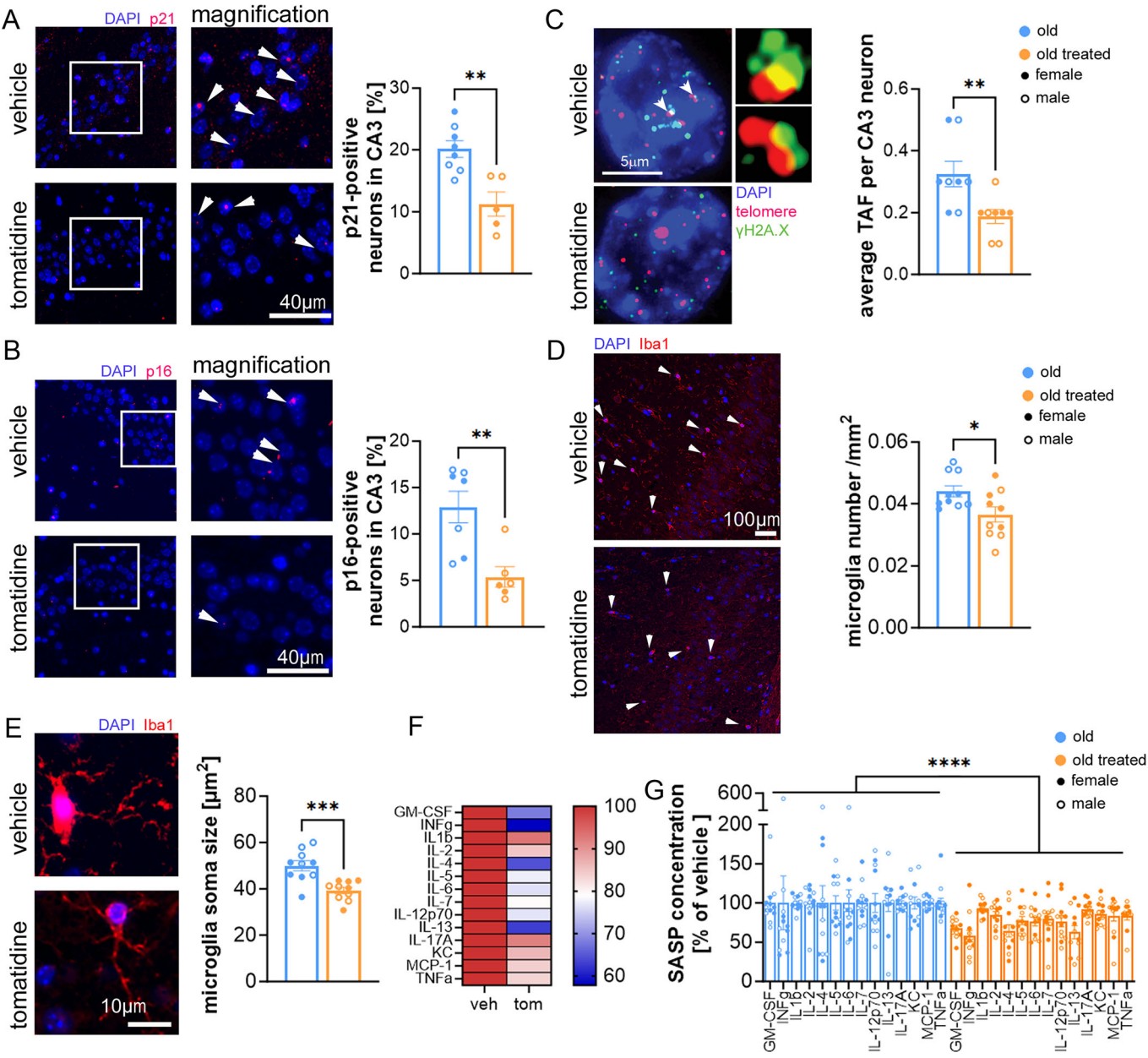

**Figure 3. Neuronal and microglial measures in the aged hippocampus following tomatidine treatment.**

(A) Representative RNA-ISH images showing p21 expression in CA3 hippocampal neurons (blue: DAPI; red: p21; scale bar: 40 μm) and corresponding quantification of p21-positive cells ($p = 0.0012$); white arrows indicate positive cells. (B) Representative RNA-ISH images showing p16 expression in CA3 hippocampal neurons (blue: DAPI; red: p16; scale bar: 40 μm) and corresponding quantification of p16-positive neurons in the CA3 region ($p = 0.0022$; white arrows indicate positive cells). (C) Quantification of telomere-associated DNA damage foci (TAF) per neuron in CA3 ($p = 0.0055$); white arrows indicate colocalization of telomeres and γ-H2AX (TAF). (D) Representative Iba1 immunofluorescence images of microglia in the hippocampus (blue: DAPI; red: Iba1; scale bar: 100 μm) and corresponding quantification of microglial cell number ($p = 0.0115$), white arrows indicate iba-1 positive cells. (E) Representative images illustrating microglial soma size (blue: DAPI; red: Iba1; scale bar: 10 μm) and corresponding quantification ($p = 0.0003$). (F) Heatmap showing mean cytokine expression levels in hippocampal homogenates. (G) Individual cytokine expression values were displayed as bar graphs for each mouse ($p < 0.0001$). Data were presented as mean ± s.e.m. Statistical significance was determined using an unpaired *t*-test or nested *t*-test in (I). $n = 5$–12, biological replicates. Males are represented by open circles, females by filled circles. Source data are available online for this figure.

## Tomatidine reduces brain endothelial cell senescence and increases tight junction gene expression

BBB dysfunction is a well-established feature of aging and has been linked to increased permeability and altered expression of tight junction proteins (Andjelkovic et al, 2023; Zhao et al, 2020). Recent work has shown that aging-associated BBB disruption is, at least in part, mediated by BEC senescence, suggesting that senescent endothelial cells contribute to the transition from a young to an aged BBB phenotype (Novo et al, 2024). Based on this, we sought to determine whether tomatidine could modulate BEC senescence in aged mice.

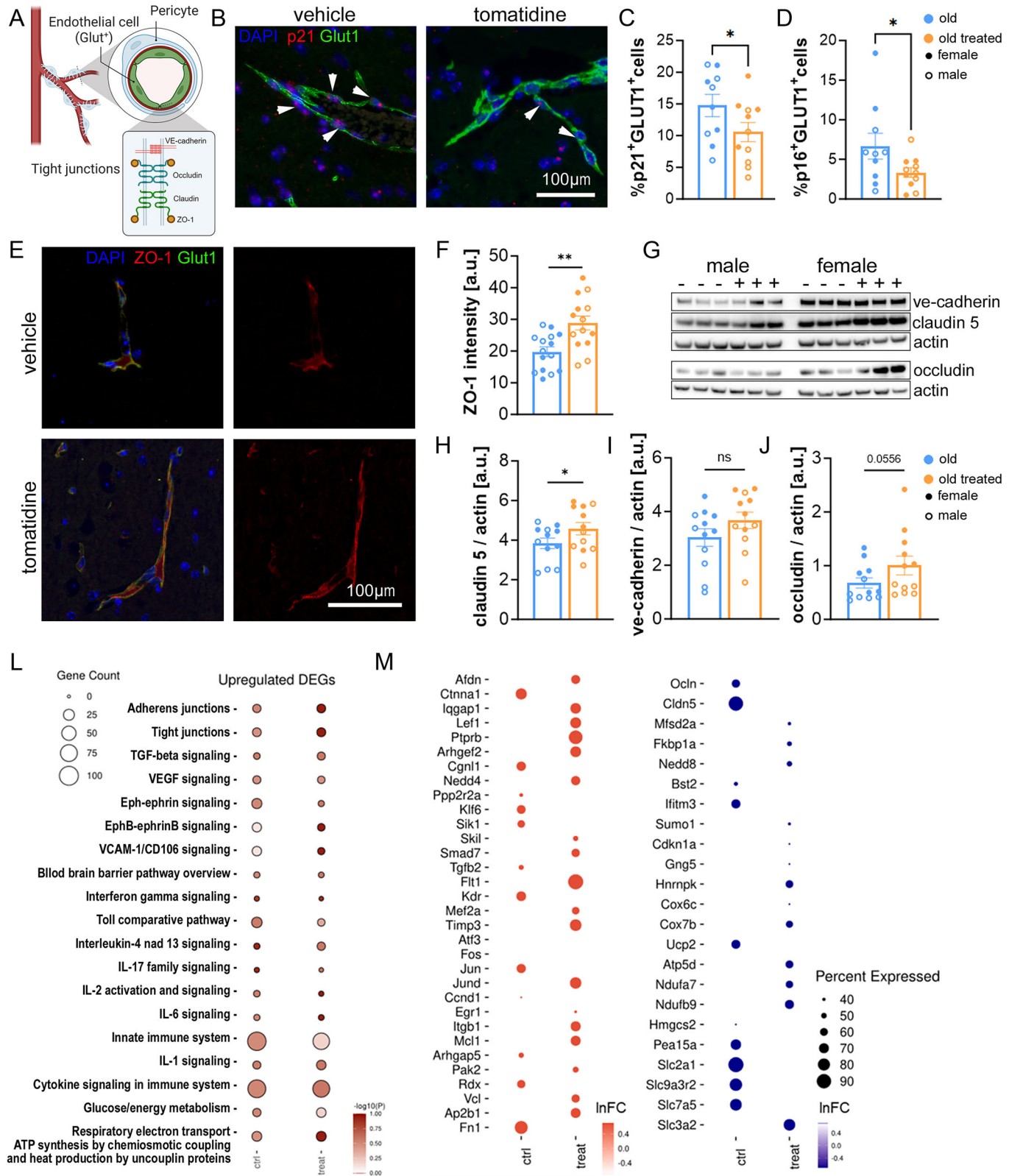

**Figure 4. Endothelial cell markers and tight junction-related measurements in the aged mouse brain.**

(A) Schematic illustration of the cellular composition and structure of the blood–brain barrier (BBB). (B) Representative RNA-ISH images showing p21 mRNA expression in GLUT1+ brain endothelial cells (blue: DAPI; red: p21; green: GLUT1). White arrows indicate positive endothelial cells. (C) Quantification of p21+/GLUT1+ endothelial cells ($p = 0.0403$). (D) Quantification of p16+/GLUT1+ endothelial cells ($p = 0.0311$). (E) Representative immunofluorescence images showing ZO-1 staining in GLUT1+ endothelial cells (blue: DAPI; green: GLUT1; red: ZO-1). (F) Quantification of ZO-1 signal intensity in brain endothelial cells ($p = 0.0019$). (G) Representative Western blot images of VE-cadherin, claudin 5, occludin, and β-actin in hippocampal lysates. (H–J) Quantification of Western blot band intensity for (H) claudin 5 ($p = 0.0433$), (I) ve-cadherin ($p = 0.1644$), and (K) occluding ($p = 0,0556$). (K) Dot plot showing enriched signaling pathways identified by differential expression analysis of endothelial cells using single-cell RNA-seq analysis from control and tomatidine-treated mice. (L) Dot plots showing selected upregulated (left) and downregulated (right) genes associated with pathways shown in panel (K). Data were presented as mean ± s.e.m. Statistical significance was determined using an unpaired $t$-test. Scale bars = 100 μm, $n = 5$–12, biological replicates. Males are represented by open circles, females by filled circles. Source data are available online for this figure.

To address this, we identified endothelial cells using the glucose transporter Glut1, a well-established marker of BECs (Fig. 4A,B) and performed RNA-ISH to assess the expression of p16Ink4a and p21Cip1 within Glut1+ cells. Tomatidine treatment significantly reduced the number of p16+/Glut1+ and p21+/Glut1+ cells in the aged brain, indicating a reduction in BEC senescence (Figs. 4B–D and EV5A,B). Given that tight junction disruption is a hallmark consequence of BEC senescence, we next examined the expression of key tight junction proteins to determine whether tomatidine preserves BBB integrity in aged mice. Immunofluorescence showed increased expression of ZO-1, a scaffolding protein that links tight junction components to the actin cytoskeleton, in brain vessels of tomatidine-treated mice (Fig. 4E,F). Additionally, western blot analysis revealed higher levels of claudin 5 and occludin, integral membrane proteins that form the paracellular barrier of endothelial tight junctions, confirming improvements in tight junction integrity (Fig. 4G–J). To gain more insight into the role of tomatidine in BEC senescence and tight junctions, we performed single-cell RNA-seq in the hippocampus from control and tomatidine-treated mice. This analysis confirmed a reduction in p21Cip1 expression specifically in BECs, with no significant change in p16Ink4a (Fig. EV5D,E). In addition, we observed increased expression of multiple tight junction-associated genes, suggesting that tomatidine enhances BBB integrity, at least in part, through suppression of BEC senescence (Fig. 4K,L).

Altogether, these results identify tomatidine as a compound that counteracts senescence-associated decline and supports brain health during aging.

## Discussion

Tomatidine is a naturally occurring compound present in unripe tomatoes and other parts of the tomato plant, with growing interest for its nutritional and pharmacological potential. Previous studies have demonstrated that tomatidine extends lifespan and healthspan in *Caenorhabditis elegans* and improves muscle function and regeneration in aged mice, including reducing skeletal muscle atrophy, enhancing strength, and increasing exercise capacity (Ebert et al, 2015; Fang et al, 2017). Mechanistically, tomatidine has been implicated in the activation of mitophagy, inhibition of the NF-κB and MAPK signaling pathways, and suppression of ATF4 activity (Bailly, 2021; Fang et al, 2017). However, its potential to target cellular senescence, a key driver of aging and age-related dysfunction, has not been previously explored.

Our study identifies tomatidine as a compound with senother-apeutic properties, capable of modulating senescence both in vitro and in vivo. In this work, we first identified tomatidine through a high-content phenotypic screen in mouse embryonic fibroblasts, where it reduced SA-beta-Gal activity without affecting cell viability. Subsequent analyses in senescent human fibroblasts and brain endothelial cells, revealed that its predominant effect is suppression of SASP factors, without altering p16INK4a or p21CIP1 levels, consistent with a senomorphic effect. In aged mice, dietary tomatidine administered for only 3 months was sufficient to reduce frailty, improve neuromuscular coordination, and enhance cognitive performance, highlighting its potential to restore healthspan in late life.

Tomatidine reduced the expression of senescence-associated markers p16Ink4a, p21Cip1, and TAF in multiple tissues, including liver, skin, hippocampal neurons, and brain endothelial cells. In the brain, these changes were accompanied by a reduction in microglial activation and inflammatory cytokines. This broader in vivo reduction suggests that tomatidine may influence senescence not only by suppressing the SASP, but also by limiting the bystander effect associated with senescent cells, thereby reducing the spread of senescence to neighboring cells or enhancing the removal of existing senescent cells.

Importantly, we demonstrate that tomatidine reduces senescence-associated markers in brain endothelial cells (BECs) and increases the expression of tight junction-associated genes. As BEC senescence has been linked to BBB dysfunction during aging (Novo et al, 2024), these findings raise the possibility that tomatidine helps preserve BBB integrity. However, a limitation of our study is that BBB permeability was not directly assessed, preventing us from establishing a causal relationship between BEC senescence, tight junction regulation, and BBB function.

A strength of our study is the comprehensive, multi-tissue analysis of senescence markers and functional outcomes following a short-term intervention in aged animals, supporting tomatidine's potential for translational aging therapies. The observation that tomatidine improves both systemic and cognitive aging phenotypes further supports its broad gerotherapeutic potential.

Nonetheless, important questions remain. The precise molecular targets of tomatidine in senescent cells have not yet been defined, and its actions are likely pleiotropic. While our findings strongly suggest a role in suppressing the SASP and reducing senescent cell burden, further work is needed to determine whether these effects in vivo are mediated by enhanced immune-mediated clearance, inhibition of senescent cell survival pathways, or direct modulation of the SASP.

In summary, our work identifies a previously unrecognized senescence-modulating role for tomatidine, a natural and orally administered compound. Across multiple tissues, including the

brain, tomatidine reduced markers of senescence and inflammation and improved functional outcomes in aged mice. While comprehensive toxicological and long-term safety studies remain to be conducted, these findings highlight tomatidine as a promising candidate for future mechanistic and translational studies aimed at improving health and cognitive function during aging.

# Methods

### Reagents and tools table

| Reagent/resource | Reference or source | Identifier or catalog number |
|---|---|---|
| **Experimental models** | | |
| C57BL/6 (*M.musculus*) | Jackson Laboratory | RRID:IMSR_JAX:000664 |
| **Recombinant DNA** | | |
| N/A | | |
| **Antibodies** | | |
| Actin | Thermo Fisher | A2066 |
| Claudin-5 | Invitrogen | 34-1600 |
| γ-H2AX | Cell Signalling | 9718S |
| GLUT1 | Abcam | ab115730 |
| IBA1 | FujiFilm | 019-19741 |
| Occludin | Abcam | AB216327 |
| VE-Cadherin | R&D systems | #AF1002 |
| ZO-1 | Invitrogen | 33-9100 |
| **Oligonucleotides and other sequence-based reagents** | | |
| *CDKN1A* | IDT | Hs.PT.58.40874346.g |
| *CDKN2A* | IDT | Hs.PT.58.40743463.g |
| CCL2 | IDT | Hs.PT.58.45467977 |
| IL1α | IDT | Hs.PT.58.40913627 |
| IL1β | IDT | Hs.PT.58.1518186 |
| IL6 | IDT | Hs.PT.58.24382464.g |
| IL8 | IDT | Hs.PT.58.39926886.g |
| TBP | IDT | Hs.PT.39a.22214825 |
| Telomere-specific TelC-Cy3 peptide nucleic acid probe | Panagene | F1002 |
| RNA-ISH probes | ACDBio | 411011 (p16) \| 408551 (p21) |
| **Chemicals, enzymes and other reagents** | | |
| Tomatidine | Enzo Life Sciences | BML-GR335-0000 |
| SA-β-gal | Thermo Fisher Scientific | C12FDG |
| Hoechst 33342 | Thermo Fisher Scientific | H1399 |
| ProLong™ Gold Antifade Mountant containing DAPI | Invitrogen | P36935 |
| Power SYBR® Green PCR Master Mix | Invitrogen | 4367659 |
| PerfeCTa ToughMix | QuantaBio | 95112–250 |
| Bio-Rad protein assay-reagent A | Bio-Rad | 500-0113 |

| Reagent/resource | Reference or source | Identifier or catalog number |
|---|---|---|
| Bio-Rad protein assay-reagent B | Bio-Rad | 500-0114 |
| Bio-Rad protein assay-reagent C | Bio-Rad | 500-0115 |
| Clarity ECL Western Blot Substrate | Bio-Rad | 170–5060 |
| KwikQuant Western blot detection kit | Kindle Bioscience | R1100 |
| Citrate buffer | Agilent-Dako | S236984 |
| **Software** | | |
| EthoVision XT | Noldus Information Technology, Wageningen, Netherlands | |
| GraphPad Prism version 8 | GraphPad Software, San Diego, CA | |
| Med-PC Software Version 4.0 | Med Associates | |
| Bio-Rad CFX Manager software | Bio-Rad | |
| Quantity One software | Bio-Rad | |
| **Other** | | |
| Mouse Cytokine Array/Chemokine Array18-Plex | Eve Technologies: Canada | MDHSTC18 |
| RNA-ISH kit | ACDBio | 322350 |
| RotaRod | Ugo Basile | 47650 |
| Real-Time System | Bio-Rad | CFX96™ |
| Thermal Cycler | Bio-Rad | C100™ |
| iBright System | Invitrogen | 1500 |
| Chromium Next GEM Single-Cell Fixed RNA Preparation Kit | 10x Genomics | CG000553 |
| Chromium Fixed RNA Profiling Kit | 10x Genomics | CG000477 |
| AVITI 2×75 High Output Cloudbreak Freestyle Kit | Element Biosciences | 860-00015 |
| Avidin Biotin Blocking Kit | Vector Labs | SP-2001 |
| Microscope | Leica | DFC7000GT |

### Experimental model

Experimental procedures were approved by the Institutional Animal Care and Use Committee (IACUC) at Mayo Clinic (protocol A00005445). C57BL/6 wild-type mice (both sexes) were purchased from Jackson Laboratory (Bar Harbor, ME, USA) at 20 months of age and housed in a pathogen-free facility under controlled environmental conditions, including a temperature range of 23–24 °C and a 12-h light/dark cycle. Mice were maintained in static, autoclaved HEPA-ventilated microisolator cages (27 × 6.5 × 15.5 cm) containing autoclaved Enrich-o'Cobs bedding. Bedding and cages were changed biweekly in class II

biosafety cabinets. Routine quarterly pathogen testing consistently yielded negative results. At 21 months of age, mice were randomly assigned to one of two dietary groups. Animals received ad libitum access to water and either standard mouse chow (Lab Diet 5053, St. Louis, MO) or chow supplemented with 0.05% tomatidine (Diets Inc., New Jersey, USA). Tomatidine (BML-GR335-0000; ≥90% purity by thin-layer chromatography, TLC) was purchased from Enzo Life Sciences (Farmingdale, NY, USA) and incorporated into the diet by Research Diets Inc. (New Brunswick, NJ, USA). The dietary intervention was maintained for 3 months. For comparison, we included a separate cohort of 4-month-old young mice (C57BL/6) of both sexes, also obtained from Jackson Laboratory, which served as untreated controls. At the end of the treatment period, mice were euthanized, and organs were harvested for downstream analyses. Mouse tissues were either snap-frozen in liquid nitrogen for biochemical analysis or fixed in 4% paraformaldehyde (PFA) for 24 h. Fixed samples were then processed, embedded in paraffin, and sectioned at 3-μm thickness for further analysis.

## Senotherapeutic screening

Senescence was evaluated based on SA-β-gal activity using a C12FDG staining assay. Senescent $Ercc1^{-/-}$ MEFs were generated by passaging the cells three times at 20% $O_2$ and then seeded at 3000 cells per well in black-wall, clear-bottom 96-well plates at least 16 h prior to treatment. Following the addition of compounds, cells were incubated for 48 h at 20% $O_2$. After removing the medium, cells were incubated with 100 nM Bafilomycin A1 in culture medium for 60 min to induce lysosomal alkalinization, followed by 20 μM C12FDG (7188, Setareh Biotech, USA) for 2 h and counterstaining with 2 μg/mL Hoechst 33342 (H1399, Thermo Fisher Scientific, MA, USA) for 15 min. Cells were then washed with PBS, fixed in 2% paraformaldehyde for 15 min, and imaged with six fields per well using a Cytation 1 high-content fluorescence imaging system (BioTek, VT, USA).

## Y maze

Spatial and working memory assessments in mice were conducted using a Y maze setup. The Y maze apparatus consists of three identical arms with elevated walls. Mice were habituated to the testing room for at least one hour prior to the experiment. During the training phase, one arm (the novel arm) was blocked, and mice explored the other two arms for 10 min. Following a 30-min rest period, the test phase was conducted, during which all three arms were accessible for 5 min. Behavior was recorded and analyzed using EthoVision XT (Noldus, Wageningen, Netherlands). Cognitive performance was evaluated by measuring spontaneous alternation, specifically the number of entries into the novel arm and the total time spent in it.

## Stone T maze

Cognitive performance was assessed using a water-motivated version of the Stone T maze, custom-built by the Mayo Clinic workshop, which is sensitive to age-related deficits in learning and memory. The apparatus consisted of an acrylic-roofed maze positioned in a shallow steel pan filled with water (~3 cm),

motivating mice to walk and to navigate toward a dry goal box. On Day 1, mice underwent straight-run training to acclimate to the escape paradigm. Each mouse was placed in the start box and allowed to traverse a straight maze to reach the goal box. Upon entry, mice were gently dried with a towel and placed in a heated holding cage. The testing room was maintained at a comfortable ambient temperature, and the holding cage was partially warmed with a heat lamp, allowing self-regulated exposure to heat. Each mouse completed three straight-run trials per session, with three sessions separated by 15-min intervals, totaling nine training trials. Maze testing was conducted the following day. After one hour of habituation in the testing room, each mouse was placed in the start box and allowed to explore the maze for up to 6 min. Latency to reach the goal box and the number of errors defined as full-body entries into incorrect arms, including each distinct path at T-junction dead ends were recorded. Inter-trial intervals of 15 min were maintained. After each trial, mice were dried with a towel and returned to the heated holding cage. Trials in which a mouse failed to reach the goal box within 5 min were marked as unsuccessful. Mice exhibiting three consecutive unsuccessful trials or signs of impaired thermoregulation (e.g., excessive shaking) were excluded from further testing. Each mouse completed a total of nine acquisition trials.

## Elevated plus maze

Anxiety-like behavior and locomotor activity were assessed using the elevated plus maze (EPM) test. The apparatus consisted of two open arms (25 × 5 cm each) and two closed arms (25 × 5 cm each), arranged at right angles to a central platform (5 × 5 cm) and elevated 40 cm above the floor. To minimize environmental stress, mice were acclimatized to the testing room for at least 1 h prior to the experiment. Each mouse was placed individually at the center of the maze, facing an open arm, and allowed to explore freely for 5 min. Behavior was recorded using a video camera and analyzed with EthoVision XT tracking software (Noldus Information Technology, Wageningen, Netherlands). Anxiety-like behavior was evaluated by quantifying the number of entries into open arms and the total time spent in them.

## RotaRod

Motor coordination and maximal walking speed were assessed using an accelerating RotaRod (Ugo Basile, RotaRod 47650). Mice were trained for three consecutive days prior to testing. Training sessions consisted of mice remaining on the rotating rod at constant speeds of 4, 6, and 8 revolutions per minute (rpm) for up to 300 s on days 1, 2, and 3, respectively. On the test day, the rod accelerated linearly from 4 to 40 rpm over 300 s. The latency to fall and the corresponding speed at the time of fall were recorded. Each mouse completed three test trials, and the average performance across trials was used for analysis.

## Pole test

Balance and motor coordination were assessed using the Pole Test. Mice were placed at the center of a horizontal pole (60 cm in length), elevated 50 cm above ground. To ensure animal safety, a box filled with

soft padding was positioned beneath the apparatus to cushion potential falls. During each trial, mice were allowed to move freely in either direction along the pole for up to 60 s. A trial was considered successful if the mouse remained on the pole for the entire duration without falling. Each mouse completed five trials, with 30-s rest intervals between trials. Time spent on the pole during each trial was recorded as a measure of motor coordination and balance.

## Forelimb grip strength analysis

Forelimb grip strength was measured using a grip strength meter equipped with a computer-integrated force transducer. Mice were allowed to grasp a horizontal metal bar with their forelimbs and were then gently pulled backward until they released the bar. The peak force exerted during each attempt was recorded. Each mouse completed three consecutive trials, and the average peak force was used for analysis.

## Frailty measurements

Frailty was evaluated using a 27-parameter clinical frailty index, as previously described by (Whitehead et al, 2013). Each parameter was scored on a three-point scale: 0 for absent, 0.5 for mild, and 1 for severe phenotype. Assessed parameters included evaluations of the integument, musculoskeletal system, vestibulocochlear/auditory function, ocular and nasal systems, digestive system, urogenital system, respiratory system, signs of discomfort, as well as measurements of body weight (g) and body surface temperature (°C). This index provides a non-invasive, quantitative measure of overall health status in aging mice. The frailty index was calculated as the average score across all parameters assessed. We measured body weight, body temperature, fore limb strength but these measures are not included in the combined frailty index.

## Human IMR90 lung fibroblasts

Human IMR90 lung fibroblasts were obtained from American Type Culture Collection (ATCC) and cultured in EMEM medium with 10% FBS and pen/strep antibiotics. Cells were obtained from an authenticated commercial source and were used without further independent authentication. Mycoplasma testing is performed regularly. To induce senescence, cells were treated with 20 Gy X-ray irradiation.

## Human brain endothelial cells (HBECs)

Human brain endothelial cells (HBECs, #1000) were cultured in Endothelial Cell Medium (ScienCell, #1001) supplemented with FBS (#0025), endothelial cell growth supplement (#1052) and penicillin/streptomycin (#0503). Cells were obtained from an authenticated commercial source and were used without further independent authentication. Mycoplasma testing is performed regularly. Cells were maintained at 37 °C in a humidified incubator with 3% $O_2$ and 5% $CO_2$. To induce senescence, cells were treated with 10 Gy X-ray irradiation.

### Senolytic assay of tomatidine

On day 10 post-irradiation, when cells exhibited a fully senescent phenotype, cultures were treated with tomatidine hydrochloride at the indicated concentrations for 48 h. Vehicle-treated controls received 0.1% DMSO.

### Senomorphic assay of tomatidine

Immediately after irradiation, cells were cultured in medium containing tomatidine, which was replenished every other day for 10 days. On day 10, cells were harvested for RNA extraction and quantitative PCR to assess changes in senescence-associated and SASP gene expression. Non-irradiated cells cultured under identical conditions served as controls.

## RT-qPCR

Total RNA was isolated from homogenized tissue biopsies or cells using TRI Reagent (Sigma-Aldrich) or the RNeasy Mini Kit (Qiagen, 74106), following the manufacturer's instructions. Complementary DNA (cDNA) was synthesized using the High-Capacity cDNA Reverse Transcription Kit (Thermo Fisher, 4368814) according to the manufacturer's protocol. Quantitative real-time PCR (RT-qPCR) was performed using one of the following reagent kits: Power SYBR® Green PCR Master Mix (Invitrogen, 4367659) on a C100™ Thermal Cycler (Bio-Rad), PerfeCTa ToughMix (QuantaBio, 95112–250) on a CFX96™ Real-Time System (Bio-Rad), or Brilliant III Ultra-Fast SYBR Green qPCR Master Mix (Agilent Technologies) on a Bio-Rad Real-Time PCR System. Data acquisition and analysis were done using Bio-Rad CFX Manager software. Predesigned primers and probes from IDT PrimeTime.

## Western blotting

Previously prepared hippocampal homogenates were used for protein analysis. Protein concentrations were determined using the Bio-Rad protein assay (Bio-Rad, reagent A, 500-0113; reagent B, 500-0114; reagent C, 500-0115). Equal amounts of protein from each sample were resolved on Tris-glycine gels, and samples were then blotted onto a 0.45 μm polyvinylidene difluoride (PVDF) membrane (Millipore) using Trans-Blot SD Semi-Dry Transfer Cells (Bio-Rad). Membranes were blocked with 5% milk powder in TBS-T (TBS with 0.05% Tween-20 in PBS) at room temperature for 1 h and then incubated with primary antibodies in 1x TBS-T with 5% BSA overnight. The primary antibodies used were rabbit anti-claudin 5 (1:1000; 34-1600, Invitrogen), rabbit anti-occludin (1:1000; AB216327, Abcam), goat anti-ve-cadherin (1:1000; #AF1002, R&D systems), and rabbit anti-actin (1:1000; A2066. Thermo Fisher). After washing in TBS-T, the membranes were incubated with a peroxidase-conjugated secondary antibody in a dilution of 1:5000 for 1 h at room temperature. The membranes were then incubated with either Clarity ECL Western Blot Substrate (Bio-Rad, 170–5060) or the KwikQuant Western blot detection kit (Kindle Bioscience, R1100) according to the manufacturer's instructions and visualized using iBright 1500 (Invitrogen). The optical density of immunoreactive bands was quantified using Quantity One software (Bio-Rad, California, USA). Protein expression levels were normalized to β-Actin and reported as relative values compared to the control group.

## Single-cell RNA-sequencing

Fixed single-cell suspensions were prepared from flash-frozen mouse brain tissue using the Chromium Next GEM Single-Cell Fixed RNA Preparation Kit (10x Genomics, CG000553, Rev B), following the manufacturer's protocol. Tissue dissociation was

performed with the gentleMACS Octo Dissociator (Miltenyi Biotec). Library construction was carried out using the Chromium Fixed RNA Profiling Kit (10x Genomics, CG000477, Rev C) in a single-plex format. Probe hybridization was conducted for 20 h at 42 °C, and indexing PCR was performed for 12 cycles. Prepared libraries were sequenced on the AVITI Sequencing Platform (Element Biosciences) using the AVITI 2 × 75 High Output Cloudbreak Freestyle Kit (Cat. No. 860-00015, Element Biosciences.

Sequencing data were aligned to the murine reference genome mm10. Data with at least 500 unique molecular identifiers (UMIs), log10 genes per UMI >0.8, >250 genes per cell, and a mitochondrial ratio of less than 20% were extracted, normalized, and integrated using the Seurat package v5.2.1 in R4.4.2. After quality control and integration, we performed a modularity-optimized Louvain clustering, leading to 13 distinct clusters. Subsequently, we performed the labeling for these 13 clusters manually with established key marker genes. The dotplot was designed using the Seurat package in accordance with the ggplot2 v3.5.2 package. The feature plots were created with the Seurat package, and the percentage of p16+ and p21+ cells was calculated according to any normalized count within each cell.

## QC sequencing

Initial QC was performed using CellRanger summary metrics for both the control and treated samples. The control library contained 15,912 cells, and the treatment library contained 15,185 cells, with high-confidence read mapping in both datasets (ctrl: 84.68%, treat: 84.10%). Median per-cell sequencing depth and complexity were similar between samples: The control cells showed a median read per cell of 13,191 (median UMI/cell 7557, median detected genes/cell 3498), while the treatment cells showed a median read per cell of 13,347 (median UMI/cell 7.891, median detected genes/cell 3690).

Cells with low library complexity were removed using standard thresholds (nFeature_RNA <200 or > 6000; nCount_RNA <500). Cells with >10–15% mitochondrial transcripts were discarded. Doublets were removed using DoubletFinder with sample-specific expected doublet rates (1–2% for ~15k cells). Only high-confidence singlets were retained. After QC filtering, both samples retained a comparable number of high-quality cells suitable for downstream integration and analysis, a batch correction was performed with Seurat v5 (RPCA-based) removing technical effects. After integration, UMAP was inspected to confirm successful mitigation of batch effects.

### QC p21

To evaluate whether p21 expression could serve as a screening-quality readout across cell types, we computed classical assay metrics including the Z-factor, signal-to-background ratio (S/B), and coefficients of variation for p21-positive and p21-negative populations within each cluster. As expected for single-gene raw scRNA-seq counts, the p21-negative population exhibited uniformly zero expression across all cell types ($\mu = 0$, $\sigma = 0$), while p21-positive cells showed moderate expression levels ($\mu = 0.45$–$1.29$) but substantial variability (CV = 28–91%). This one-sided and zero-inflated distribution caused all S/B values to be infinite and resulted in strongly negative Z-factors in nearly all clusters (e.g., Interneurons: Z = −1.20; Neurons: Z = −0.96; Astrocytes: Z = −0.91; Microglia: Z = −0.74). Only rare cell types with extremely small positive subsets, such as macrophages ($n = 4$),

produced slightly less negative or near-zero Z values (Z ≈ 0.16), but these estimates are not meaningful due to the small sample size. Collectively, these results confirm that raw p21 expression is unsuitable as a screening-quality assay readout in scRNA-seq data, as the inherent sparsity, zero inflation, and high variance among positive cells violate the assumptions underlying classical high-throughput screening metrics.

### QC p16

We next computed classical screening-performance metrics (Z-factor, S/B ratio, and CV) for p16 expression across all annotated brain cell types. As with p21, the p16-negative population showed uniformly zero expression in every cluster ($\mu = 0$, $\sigma = 0$), whereas p16-positive cells displayed moderate expression levels ($\mu = 0.12$–$1.49$) but substantial variability (CV ranging from 28 to 130%). Because the negative population exhibited no measurable signal or variance, all S/B ratios were infinite, and all Z-factors were strongly negative across cell types (e.g., Interneurons: Z = −1.33; Endothelial cells: Z = −2.91; Neurons: Z = −0.68; Astrocytes: Z = −0.59). Only very small p16+ subsets (e.g., choroid plexus cells: $n = 2$) produced marginally positive or near-zero Z-values (Z ≈ 0.16), but these estimates are not meaningful due to extremely low sample size. The consistently negative Z-factors reflect the inherently zero-inflated, one-sided nature of p16 transcription in single-cell RNA-seq data, where all negative cells have absolute zero counts and positive cells show high stochastic variability. Consequently, raw p16 expression is unsuitable as a high-content screening readout, and classical assay metrics such as Z and S/B do not provide interpretable assay-quality information for this marker in scRNA-seq. BBB We used the paper by (Zhao et al, 2020) as a framework to generate the graphs in Figure 4l, m.

## Cytokine array

Cytokine levels in hippocampal homogenates were quantified using a multiplex immunoassay platform. Tissue samples were homogenized in lysis buffer containing protease inhibitors and centrifuged to remove debris. Supernatants were collected, and total protein concentration was determined using the Bio-Rad protein assay. Samples were then submitted to Eve Technologies Corporation (Calgary, Alberta, Canada) for analysis using the Mouse Cytokine/Chemokine 18-Plex Discovery Assay® (MDHSTC18) platform. The assay was performed according to the company's standard protocols. Cytokine concentrations were calculated from standard curves and are expressed in pg/mL. All values were normalized to total protein content and reported relative to control groups.

## Tissue section immunostaining

Formalin-fixed, paraffin-embedded (FFPE) tissue sections (3 μm) were deparaffinized in HistoClear and rehydrated through a graded ethanol series: 100% ethanol, 90% ethanol, and 70% ethanol, followed by two washes in distilled water. Antigen retrieval was performed by boiling slides in citrate buffer (pH 6.0; Agilent-Dako, S236984) for 10 min, followed by cooling at room temperature for 20 min and two PBS washes (5 min each). Sections were blocked in 1% bovine serum albumin (BSA), 1:60 normal goat serum or fetal bovine serum in PBS for 30 min at

room temperature, then incubated overnight at 4 °C with the following primary antibodies: rabbit anti-Iba1, rabbit anti-GLUT1 and mouse anti-ZO-1. After washing with PBS, sections were incubated with species-specific secondary antibodies for 1 h at room temperature. Following three PBS washes, slides were mounted with ProLong™ Gold Antifade Mountant containing DAPI (Invitrogen) for nuclear counterstaining. For all immunohistochemical analyses, including RNA-ISH and Immuno-FISH, quantification parameters depended on the specific marker assessed. For tissue-based senescence markers (e.g., p16, p21, and TAF), a minimum of 100 cells per animal were analyzed. The number of fields examined was kept consistent across animals and experimental groups for each staining. The total number of images acquired per animal depended on the magnification used during imaging (e.g., 20× or 40× for general tissue analysis and 63× for TAF quantification).

## RNA-ISH

RNA in situ hybridization was performed using the RNAscope® 2.5 HD Reagent Kit-RED (Advanced Cell Diagnostics, ACD) according to the manufacturer's protocol. Formalin-fixed, paraffin-embedded (FFPE) tissue sections were deparaffinized in HistoClear (2 × 5 min), rehydrated in 100% ethanol (2 × 1 min), and air-dried. Sections were incubated with hydrogen peroxide ($H_2O_2$) for 10 min at room temperature, followed by two washes in distilled water. Antigen retrieval was performed by boiling slides in 1X RNAscope Target Retrieval solution for 15 min. After rinsing in distilled water and 100% ethanol, slides were air-dried and incubated with Protease Plus reagent for 30 min at 40 °C. Tissue sections were then hybridized with target-specific probes against p21 (ACD #408551) and p16 (ACD #411011) for 2 h at 40 °C. Signal amplification was carried out sequentially using AMP1 (30 min at 40 °C), AMP2 (15 min at 40 °C), AMP3 (30 min at 40 °C), AMP4 (15 min at 40 °C), AMP5 (30 min at room temperature), and AMP6 (15 min at room temperature), with ACD wash buffer (WB) used for intermediate washes between steps. Chromogenic detection was performed using the RNAscope 2.5 HD Reagent Kit-RED. Following five final washes in distilled water, slides were counterstained and mounted using ProLong™ Gold Antifade Mountant with DAPI (Invitrogen). RNA-ISH quantification was performed by manually scoring positive cells relative to the total number of cells in each field. Nuclei were first identified and counted based on DAPI staining to determine the total cell number. Cells were classified as RNA-ISH–positive if they exhibited one or more foci corresponding to p16 or p21 transcripts. The percentage of RNA-ISH–positive cells was calculated as the ratio of positive cells to total DAPI-positive cells.

## Immuno-FISH

Formalin-fixed, paraffin-embedded (FFPE) tissue sections were deparaffinized in 100% HistoClear and rehydrated through a graded ethanol series (100, 90, and 70%; 2 × 5 min each), followed by two 5-min rinses in distilled water. Antigen retrieval was performed by immersing slides in 0.01 M citrate buffer (pH 6.0) and heating to boiling for 10 min. After cooling to room temperature, sections were rinsed in distilled water for 5 min.

Tissue sections were blocked using normal goat serum (1:60 in PBS with 1% BSA) for 30 min at room temperature, followed by additional blocking with an Avidin/Biotin Blocking Kit (Vector Laboratories, Burlingame, CA) for 15 min per step. Sections were then incubated overnight at 4 °C with a rabbit monoclonal anti-γH2AX antibody (1:200; Cell Signaling Technology, #9718. Following three PBS washes, sections were incubated with a biotinylated goat anti-rabbit secondary antibody (1:200; Vector Laboratories, PK-6101) for 30 min at room temperature. After three more PBS washes, fluorescein-conjugated Streptavidin-Cy5 (1:500; Vector Laboratories, A-2011) was applied for 30 min at room temperature. Slides were washed again in PBS (3 × 5 min), then post-fixed with 4% paraformaldehyde in PBS for 20 min to stabilize antibody complexes. Sections were subsequently dehydrated in a cold ethanol gradient (70, 90, and 100%; 3 min each) and air-dried. For telomere-specific peptide nucleic acid fluorescence in situ hybridization (PNA-FISH), 10 µL of hybridization solution, containing 70% deionized formamide, 20 mM $MgCl_2$, 1 M Tris (pH 7.2), 5% blocking reagent, and 2.5 µg/mL Cy3-labeled (CCCTAA) telomere-specific PNA probe (PANAGENE), was applied to each section. Slides were denatured at 80 °C for 10 min and hybridized for 2 h at room temperature in the dark. Post-hybridization washes included 70% formamide in 2× SSC (10 min), followed by 2× SSC (10 min) and PBS (10 min). Slides were mounted with ProLong™ Gold Antifade Mountant with DAPI (Invitrogen) and imaged using high-resolution z-stacked fluorescence microscopy with a 63× objective lens. For telomere-associated foci (TAF) quantification, z-stack images were acquired to visualize DNA damage foci in three dimensions (as we have done before (Hewitt et al, 2012)). TAF were manually quantified by first identifying γ-H2AX foci within each nucleus, followed by detection of telomeric signals using a telomere-specific probe. A TAF was identified when a γ-H2AX foci colocalized with a telomeric signal throughout the z-stack. The total number of TAF per nucleus was subsequently recorded.

## Microscopic imaging

Microscopic imaging was performed using a Leica DFC7000GT inverted microscope.

## Quantification and statistical analysis

All results are expressed as mean ± standard error of the mean (SEM). Sample sizes were determined based on prior experience with similar experimental paradigms and on previously published studies, without formal power calculation. Animals were randomly assigned to experimental groups where applicable. Investigators were blinded to group allocation during data collection and analysis when feasible. Inclusion criteria were defined prior to experimentation, and no data were excluded unless technical issues (e.g., failed assays or sample loss) occurred. All statistical analyses, including the tests used, exact *n* numbers, and *p* values, are reported in the figure legends. Data distribution was assessed prior to selecting appropriate parametric or non-parametric statistical tests. Statistical analyses and graph generation were performed using GraphPad Prism version 10 (GraphPad Software, San Diego, CA). Differences between groups were assessed using unpaired two-tailed Student's *t*-tests, one-way or two-way analysis of

## The paper explained

### Problem

As we age, many cells in our tissues stop dividing and enter a state called cellular senescence. Senescent cells do not die when they should. Instead, they release a cocktail of inflammatory and tissue-damaging factors that contribute to frailty, chronic inflammation, and age-related diseases. Drugs that remove senescent cells ("senolytics") or tone down their secretions ("senomorphics") are being developed, but many of these candidates can be toxic or are still far from clinical use. There is an urgent need for safer, well-tolerated compounds that can be given later in life to reduce the harmful impact of senescent cells on health and organ function.

### Results

We systematically screened natural compounds and identified tomatidine, a plant-derived molecule, as a promising modulator of cellular senescence. In human cell models, tomatidine reduced hallmark features of senescence and lowered the production of inflammatory and tissue-remodeling factors, without forcing damaged cells to re-enter the cell cycle or causing substantial cell death. We then tested tomatidine in old mice. When given late in life, tomatidine reduced markers of senescent cells and chronic inflammation in multiple tissues and improved measures of health in these aged animals. Together, these findings show that tomatidine can selectively dampen the harmful activities of senescent cells in vitro and in vivo.

### Impact

Our work identifies tomatidine as a well-tolerated senomorphic candidate that can be given in old age to limit the detrimental effects of senescent cells on tissue function. Because tomatidine is a natural small molecule with a favorable safety profile in preclinical studies, it may be more readily translated toward clinical testing than many experimental senotherapies. These results support the broader concept that targeting senescent cells, not only by killing them but also by reprogramming their harmful secretions, could become a viable strategy to delay or alleviate multiple age-related conditions and improve healthspan in older adults.

variance (ANOVA), followed by Tukey's multiple comparisons test. A $p$ value <0.05 was considered statistically significant. Significance levels are represented as follows: $p < 0.05$ (*), $p < 0.01$ (**), $p < 0.001$ (***), and $p < 0.0001$ (****) as indicated in the respective figure legends.

## Graphics

Schematic illustrations for experimental design and organ-specific diagrams (liver, skin, and brain) in Figs. 1, 2, and 4, as well as Figs. EV1 and EV4 were created with BioRender.com.

## Data availability

The RNA-seq data generated in this study have been deposited in the Gene Expression Omnibus (GEO) under accession number GSE312732.

The source data of this paper are collected in the following database record: biostudies:S-SCDT-10_1038-S44321-026-00400-0.

## Peer review information

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

## Acknowledgements

This work was supported by NIH grants R01 AG068182 (DJ), P01 AG062413 (DJ, JFP, SK, NKL, and PDR), U54 AG079754 (PDR), U54 AG076041 (PDR), U19 AG056278 (PDR), P01 AI172501 (PDR), R01 AG086085 (SK), funding from Hevolution/AFAR (DJ), P30DK084567 (DJ), UG3/UH3CA268103 (JFP); R01AG068048 (JFP); R01AG82708 (JFP) and the Glenn Foundation for Medical Research (JFP and NKLB) FCT scholarship 2022.11293.BD (https://sciproj.ptcris.pt/90077DFA) and FLAD award (DGC). We are grateful for support from the Robert and Arlene Kogod Center on Aging for a Career Development Award (GL).

## Author contributions

**Daniela G Costa**: Formal analysis; Investigation; Methodology. **Lucy M Gee**: Formal analysis; Investigation; Methodology. **Gung Lee**: Investigation; Methodology. **Lilian Sales Gomez**: Investigation; Methodology. **Ana Catarina Franco**: Investigation. **Maria Grazia Vizioli**: Data curation; Investigation; Methodology. **Karla Valdivieso**: Formal analysis; Investigation. **Lei J Zhang**: Formal analysis; Investigation; Methodology. **Nick Pirius**: Resources; Investigation. **Helene Martini**: Investigation. **Rebecca A Poritt**: Resources; Formal analysis; Methodology. **Dominik Saul**: Resources; Software; Formal analysis; Investigation. **Claudia Cavadas**: Writing—review and editing. **Scott Ebert**: Writing—review and editing. **Nathan K LeBrasseur**: Writing—review and editing. **Sundeep Khosla**: Writing—review and editing. **João F Passos**: Writing—review and editing. **Christopher M Adams**: Writing—review and editing. **Paul D Robbins**: Funding acquisition; Writing—review and editing. **Diana Jurk**: Conceptualization; Data curation; Supervision; Funding acquisition; Writing—original draft; Writing—review and editing.

Source data underlying figure panels in this paper may have individual authorship assigned. Where available, figure panel/source data authorship is listed in the following database record: biostudies:S-SCDT-10_1038-S44321-026-00400-0.

## Disclosure and competing interests statement

CMA is an inventor on patents for the use of tomatidine in muscle health and metabolic health, which are licensed to Emmyon, Inc., where CMA and SME are shareholders and officers.

# Expanded View Figures

**Figure EV1.   Identification of tomatidine as a novel senomorphic compound.**

(A) Schematic of the senescent cell-based screening workflow. Senescent cells were generated, seeded into 96-well plates, treated with compounds, stained with C12FDG (SA-β-gal) and DAPI, imaged on a high-content system, and analyzed. (B) Concentration-response curves of tomatidine in senescent $Ercc1^{-/-}$ MEFs and in total $Ercc1^{-/-}$ MEF cultures. (C) Concentration-response curves comparing senescent $Ercc1^{-/-}$ MEFs to non-senescent WT MEFs. Responses were normalized to untreated controls and fit by nonlinear regression using a variable-slope model (GraphPad Prism). The $IC_{50}$ for senescent $Ercc1^{-/-}$ MEFs was 1.24 (95% CI: 1.03–1.46), whereas the $IC_{50}$ for non-senescent WT MEFs was 41.52 (95% CI: 21.42–140.5). The selectivity index (SI), calculated as $IC_{50}$ (WT) / $IC_{50}$ (senescent $Ercc1^{-}/^{-}$), was 33.5. Data were shown as mean ± SD from $n = 3$ independent experiments, technical replicates. "Total" denotes mixed cultures following the senescence-induction protocol. The non-senescent control data presented in panels (B, C) derive from the same experimental dataset. (D) Representative C12FDG fluorescence images (green: C12FDG/SA-β-gal; blue: Hoechst). Scale bar, 100 μm. (E) Illustration of the experimental setup for senomorphic assays. (F, G) Relative mRNA expression of p16, p21, and SASP-related genes in senescent IMR90 fibroblasts treated with tomatidine (in F) $p < 0.0001$ and in (G) IL6 $p = 0.021$ and $p = 0.005$, IL8 $p < 0.0001$ and $p < 0.0001$, IL1a $p = 0.001$ and $p = 0.031$, IL-1b $< 0.0001$ and $p < 0.0001$). (H, I) Relative mRNA expression of p16, p21, and SASP-related genes in senescent HBMECs treated with tomatidine. (for both p16 and p21 in H): $p < 0.0001$, in (I): IL-6 $p = 0.0006$ and $p < 0.0001$, IL1a $p < 0.0001$ and $p < 0.0001$, IL-1b: $p = 0.083$ and $p = 0.0008$, CCL2: $p < 0.0001$ and $p < 0.0001$). (J) Illustration of the experimental setup for senolytic assays. (K) Relative mRNA expression of p16 ($p = 0.0102$) and p21 in senescent HBMECs after 48 h of tomatidine treatment. Data were presented as mean ± s.e.m. (F–K). Statistical significance was determined using two-way ANOVA followed by Tukey's multiple comparisons test. $n = 3$, technical replicates. Source data are available online for this figure.

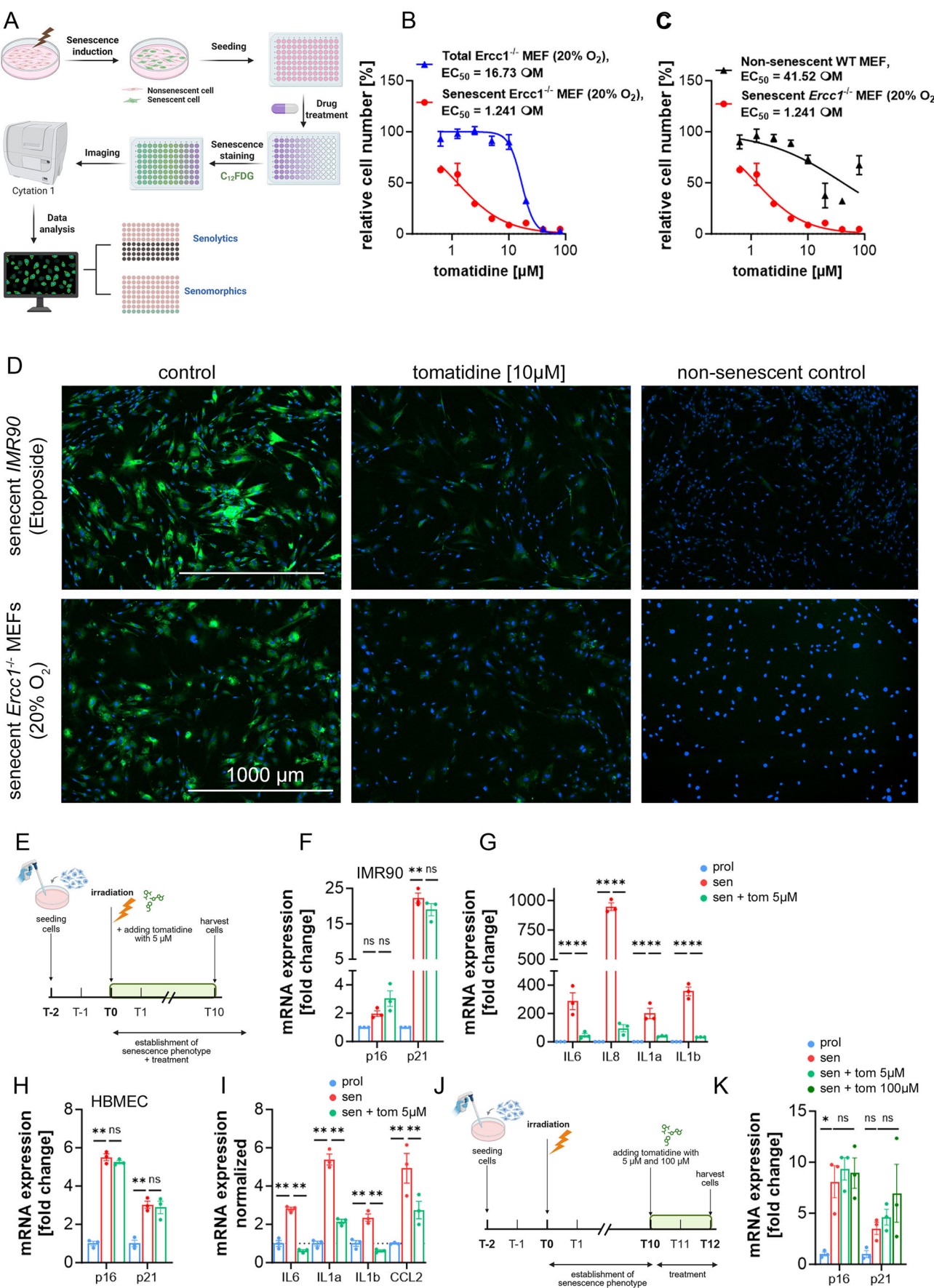

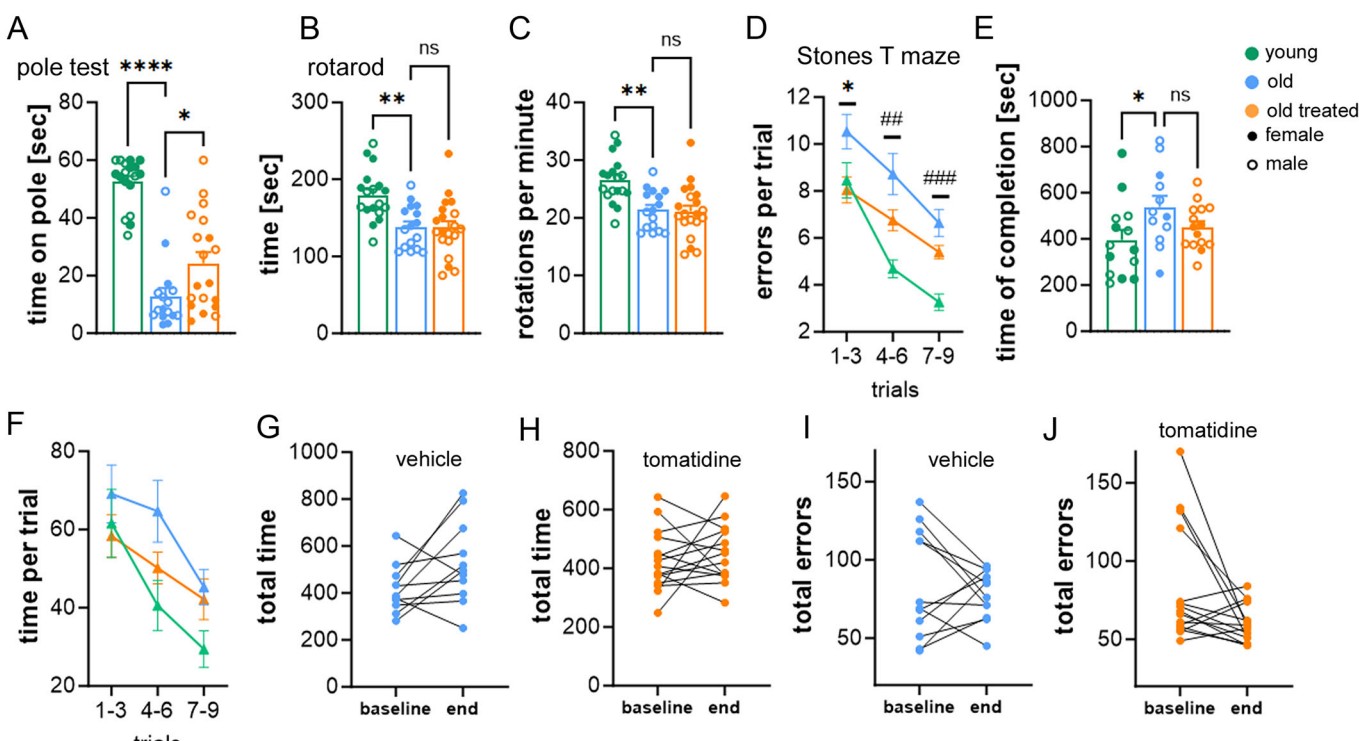

**Figure EV2. Healthspan and cognitive assessments in aged mice treated with tomatidine.**

(**A**) Performance on the pole test, measuring motor coordination and balance. (**B**, **C**) Rotarod testing showing (**B**) time spent on the rod and (**C**) maximal speed achieved. (**D**) Number of errors made in the Stone T maze across trials. Symbols: (*) indicates a significant difference between old and old tomatidine-treated groups, (##) indicates significant differences between young and old groups. Significant improvements in errors made between trial 1–3 vs 7–9 ($p = 0.0003$) and 4–6 vs 7–9 ($p = 0.0092$) in treated animals and in untreated animals, significant improvements between trial 1–3 vs 7–9 ($p = 0.0014$). (**E**, **F**) Time required to complete the Stone T maze in aged control and aged tomatidine-treated mice. (**G**, **H**) Comparison of baseline versus endpoint measurements for total time spent completing the maze. (**I**, **J**) Comparison of baseline versus endpoint measurements for total errors made prior to completing the maze. Data were presented as mean ± s.e.m. Statistical significance was determined using one-way or two-way ANOVA followed by Tukey's multiple comparisons test. $n = 15$–23, biological replicates. Males are represented by open circles, females by filled circles. Source data are available online for this figure.

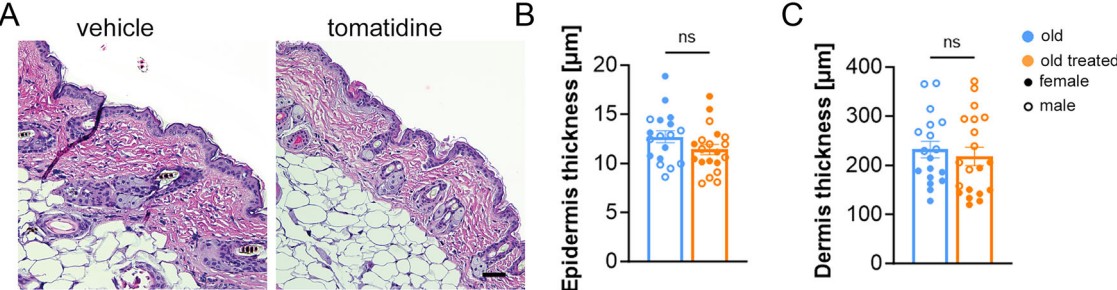

**Figure EV3. Skin thickness measurements in aged mice treated with tomatidine.**

(A) Representative H&E-stained skin sections from tomatidine-treated aged mice and untreated aged controls. (B) Quantification of epidermis thickness ($p = 0.1238$). (C) Quantification of dermal thickness ($p = 0.5839$). Data were presented as mean ± s.e.m. Statistical significance was determined using a $t$-test. Scale bar = 50 μm, $n = 18$–21, biological replicates. Males are represented by open circles, females by filled circles. Source data are available online for this figure.

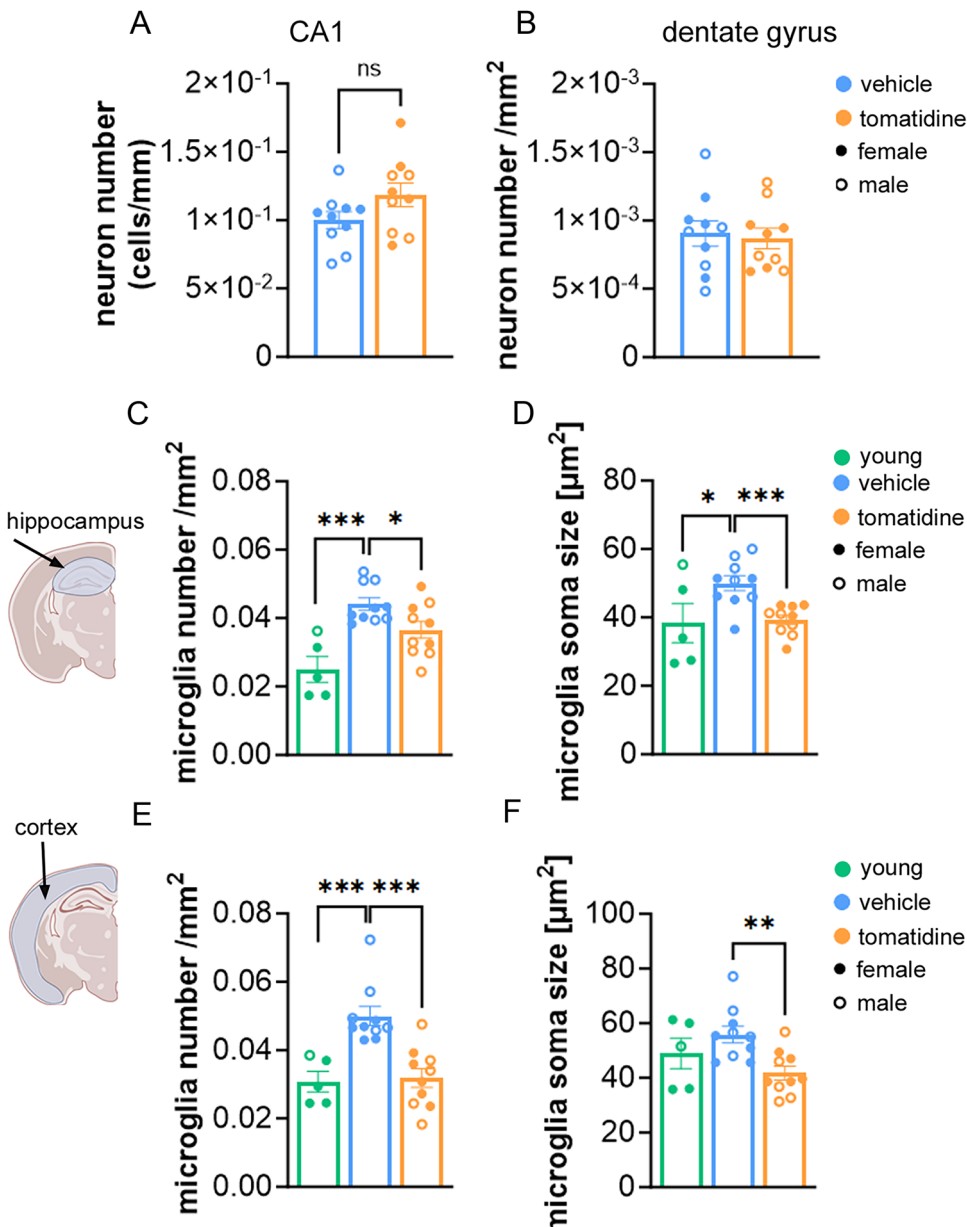

**Figure EV4. Neuronal density and microglial morphology in hippocampus and cortex of aged mice treated with tomatidine.**

(A) Quantification of neuronal density in CA1, showing the number of NeuN$^+$ neurons per mm length ($p = 0.0501$). (B) Neuronal density in the dentate gyrus (DG) region of the hippocampus ($p = 0.3802$). (C) Number of Iba1$^+$ microglia in the hippocampus; Exact $p$ values for number Iba1$^+$ (Young vs. vehicle, $p = 0.0002$; vehicle vs. tomatidine, $p = 0.0230$). (D) Microglial soma size in the hippocampus, reflecting changes in activation state; Exact $p$ values for soma size in hippocampus (young vs. vehicle, $p = 0.037$; vehicle vs. tomatidine, $p = 0.0003$). (E) Number of Iba1$^+$ microglia in the cortex; Exact $p$ values for number Iba1$^+$ (Young vs. vehicle, $p = 0.0009$; vehicle vs. tomatidine, $p = 0.0002$). (F) Microglial soma size in the cortex; Exact $p$ values for soma size in cortex (Young vs. vehicle, $p = 0.2525$; vehicle vs. tomatidine, $p = 0.0022$). Panels (C–F) present the same datasets included in the main figure; here, we additionally show the values from young control mice to allow direct comparison between young, aged, and tomatidine-treated aged groups. Data were presented as mean ± s.e.m. Statistical significance was determined using a $t$-test or one-way ANOVA. $n = 5$-10, biological replicates. Males are represented by open circles, females by filled circles. Source data are available online for this figure.

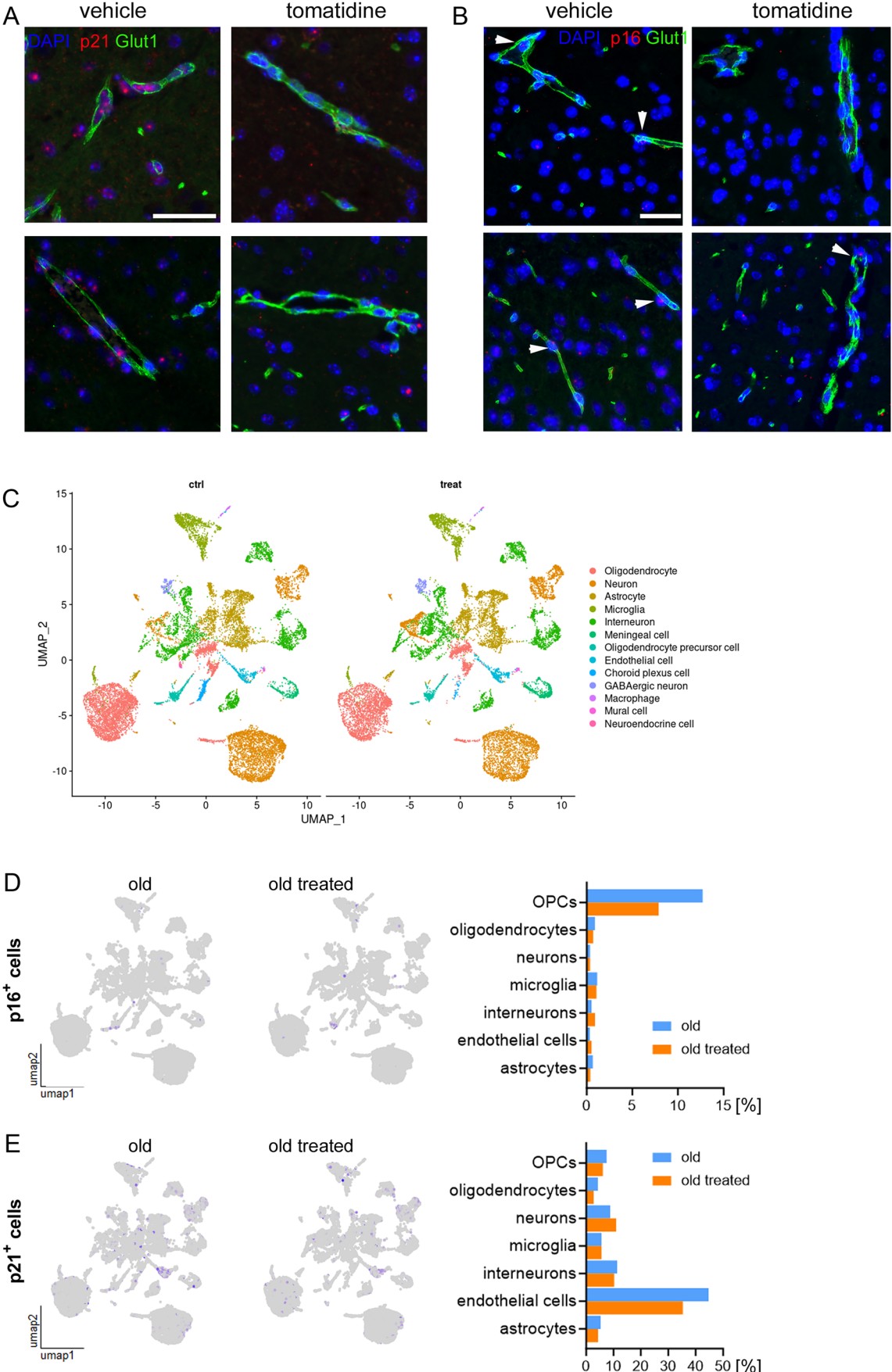

◀ **Figure EV5.  RNA-ISH and scRNA-seq analysis of p16 and p21 expression in brain endothelial cells.**

(**A**) Representative RNA-ISH images showing p21 mRNA expression in GLUT1-positive brain endothelial cells (blue: DAPI; red: p21; green: GLUT1). Scale bar 50 μm. (**B**) Representative RNA-ISH images showing p16 mRNA expression in GLUT1 positive BBB endothelial cells (blue: DAPI; red: p16; green: GLUT1). White arrows indicate positive endothelial cells. Scale bar 30 μm (**C**) UMAP representation of all cell types colored by class from tomatidine treated and untreated controls. scRNA-seq was performed on hippocampus pooled from four animals per group. (**D**) UMAP representation and quantification of p16+ cells from tomatidine treated and untreated controls. (**E**) UMAP representation and quantification of p21+ cells from tomatidine treated and untreated controls. Source data are available online for this figure.

