## [Peer Review File · EMBO Molecular Medicine]

Tomatidine is a senotherapeutic that improves cognitive function and reduces senescence in aged mice

Daniela Costa, Lucy Gee, Gung Lee, Lilian Sales Gomez, Ana Catarina Franco, Maria Vizioli, Karla Valdivieso, Lei Zhang, Nick Pirius, Helene Martini, Rebecca Poritt, Dominik Saul, Claudia Cavadas, Scott Ebert, Nathan LeBrasseur, Sundeep Khosla, Joao Passos, Christopher Adams, Paul Robbins, and Diana Jurk

Corresponding author: Diana Jurk (jurk.diana@mayo.edu)

Review Timeline:

Submission Date:	4th Nov 25
Editorial Decision:	19th Nov 25
Revision Received:	26th Dec 25
Editorial Decision:	23rd Jan 26
Revision Received:	5th Feb 26
Editorial Decision:	11th Feb 26
Revision Received:	23rd Feb 26
Accepted:	25th Feb 26

Editor: Lise Roth

Transaction Report:

19th Nov 2025

Dear Dr. Jurk,

Thank you for submitting your manuscript to EMBO Molecular Medicine. We have now received feedback from the three reviewers who agreed to evaluate it. As you will see from the reports below, while referees #1 and #2 are overall supportive of the manuscript pending minor revisions, referee #3 raises major concerns related to overstatements on tomatidine described as a safe, orally bioavailable senotherapeutic with strong translational potential to extend healthspan, and to the lack of direct mechanistic link between tomatidine and neuronal function.

Following further discussion with the referees, we would like to invite you to revise the manuscript to provide evidence supporting your claims regarding safety and bioavailability, and to strengthen the mechanistic insight. The statements on extended lifespan could either be addressed experimentally or by toning them down, given that addressing them experimentally might require an extended revision time.

Addressing the reviewers' concerns in full will be necessary for further considering the manuscript in our journal, and acceptance of the manuscript will entail a second round of review. EMBO Molecular Medicine encourages a single round of revision only and therefore, acceptance or rejection of the manuscript will depend on the completeness of your responses included in the next, final version of the manuscript. For this reason, and to save you from any frustrations in the end, I would strongly advise against returning an incomplete revision.

We are expecting your revised manuscript within three months, if you anticipate any delay, please contact us.

We require:

Additional information on source data and instruction on how to label the files are available

4) A .docx formatted letter INCLUDING the reviewers' reports and your detailed point-by-point responses to their comments. As part of the EMBO Press transparent editorial process, the point-by-point response is part of the Review Process File (RPF), which will be published alongside your paper.

5) A complete author checklist, which you can download from our author guidelines (<https://www.embopress.org/page/journal/17574684/authorguide#submissionofrevisions>). Please insert information in the checklist that is also reflected in the manuscript. The completed author checklist will also be part of the RPF.

6) All Materials and Methods need to be described in the main text using our 'Structured Methods' format. According to this format, the Methods section includes a Reagents and Tools Table (listing key reagents, experimental models, software and relevant equipment and including their sources and relevant identifiers) followed by a Methods and Protocols section describing the methods, ideally using a step-by-step protocol format. The aim is to facilitate adoption of the methodologies across labs. Please download and fill our Reagents and Tools Table template (.docx), which you can find in our author guidelines: <https://www.embopress.org/page/journal/14693178/authorguide#structuredmethods>.

7) Please note that all corresponding authors are required to supply an ORCID ID for their name upon submission of a revised manuscript.

8) It is mandatory to include a 'Data Availability' section after the Materials and Methods. Before submitting your revision, primary

datasets produced in this study need to be deposited in an appropriate public database, and the accession numbers and database listed under 'Data Availability'. Please remember to provide a reviewer password if the datasets are not yet public (see <https://www.embopress.org/page/journal/17574684/authorguide#dataavailability>).

9) For data quantification: please specify the name of the statistical test used to generate error bars and P values, the number (n) of independent experiments (specify technical or biological replicates) underlying each data point and the test used to calculate p-values in each figure legend. The figure legends should contain a basic description of n, P and the test applied. Graphs must include a description of the bars and the error bars (s.d., s.e.m.). Please provide exact p values.

10) Our journal encourages inclusion of *data citations in the reference list* to directly cite datasets that were re-used and obtained from public databases. Data citations in the article text are distinct from normal bibliographical citations and should directly link to the database records from which the data can be accessed. In the main text, data citations are formatted as follows: "Data ref: Smith et al, 2001" or "Data ref: NCBI Sequence Read Archive PRJNA342805, 2017". In the Reference list, data citations must be labeled with "[DATASET]". A data reference must provide the database name, accession number/identifiers and a resolvable link to the landing page from which the data can be accessed at the end of the reference. Further instructions are available at .

11) We replaced Supplementary Information with Expanded View (EV) Figures and Tables that are collapsible/expandable online. EV Figures should be cited as 'Figure EV1, Figure EV2' etc... in the text and their respective legends should be included in the main text after the legends of regular figures.

12) The paper explained: EMBO Molecular Medicine articles are accompanied by a summary of the articles to emphasize the major findings in the paper and their medical implications for the non-specialist reader. Please provide a draft summary of your article highlighting

13) Author contributions: CRedit has replaced the traditional author contributions section because it offers a systematic machine readable author contributions format that allows for more effective research assessment. Please remove the Authors Contributions from the manuscript and use the free text boxes beneath each contributing author's name in our system to add specific details on the author's contribution. More information is available in our guide to authors.

Please also suggest a visual abstract to illustrate your article as a PNG file 550 px wide x 300-600 px high. A cropped portion of this image will serve as thumbnail for the table of content on our webpage.

16) As part of the EMBO Publications transparent editorial process initiative (see our Editorial at <http://embomolmed.embopress.org/content/2/9/329>), EMBO Molecular Medicine will publish online a Review Process File (RPF) to accompany accepted manuscripts.

In the event of acceptance, this file will be published in conjunction with your paper and will include the anonymous referee

reports, your point-by-point response and all pertinent correspondence relating to the manuscript. Let us know whether you agree with the publication of the RPF and as here, if you want to remove or not any figures from it prior to publication. Please note that the Authors checklist will be published at the end of the RPF.

I look forward to receiving your revised manuscript.

Yours sincerely,

Lise Roth

**** Reviewer's comments ****

Referee #1 (Remarks for Author):

In this study, Costa et al screened a panel of natural compounds for their ability to reduce SA-bGal activity of peroxide-treated Ercc1-/- MEFs and identified tomatidine as a previously unrecognized compound with senescence-reducing activity. In irradiated human fibroblasts, tomatidine reduced the expression of SASP genes, but not of p21 or p16, and did not cause significant cell death, overall revealing its senomorphic activity. 24-month-old mice, treated with 0.05% tomatidine for 3 months, showed a significantly reduced frailty index, as well as improved neuromuscular function and cognitive performance. In these mice, liver and skin tissues showed a reduction in senescence markers p16, p21, and TAF, as did neurons in mouse brain. Furthermore, tomatidine also suppressed brain endothelial cell senescence and increased the expression of tight junction genes. Although this study provides few mechanistic insights into the molecular pathways affected by tomatidine, its effects on reducing the SASP of treated cells, while also reducing cellular senescence systemically, are clearly presented and convincing. Overall, this is a very well-designed and carefully executed study, revealing clear and unambiguous results and a novel senotherapeutic compound with the potential to improve health, healthspan, and cognitive function during aging.

Minor points:

- 1) How were RNA-ish data evaluated and quantified? How were TAF-positive cells quantified? These details should be included in the methods section.
- 2) It would be important to mention whether any negative/side effects were observed in mice treated for three months with tomatidine.

Referee #2 (Comments on Novelty/Model System for Author):

This paper has nicely present the senotherapeutic effect of Tomatidine in aged mice. Below are a few points that will be helpful to further enhance the study: 1. In extended figure 1D, the authors should also present the staining results of non-senescent cells as a negative control. 2. The author didn't specify the age of young mice in Figure 1, neither in the text or method. 3. In the Extended data 4 and Figure 3, it would be informative to also present the quantification of microglia density in young mice in hippocampus and cortex. This information could tell us whether aging induces microgliosis. 4. It would be helpful to also quantify the neuronal density and synaptic density changes in CA1 or dentate gyrus after drug treatment in Figure 3. 5. Figure 3E should also present a zoom out, lower mag images to show the microglia distribution/density throughout the hippocampus.

Referee #3 (Comments on Novelty/Model System for Author):

This manuscript provides a valuable contribution by presenting a relatively comprehensive screening pipeline and in vivo data,

highlighting the potential of tomatidine as a nutraceutical candidate. The authors demonstrate that tomatidine supplementation in aged mice improves cognitive performance, reduces neuroinflammation, decreases senescence-associated markers in neurons and across multiple tissues, and enhances BBB integrity. These findings are intriguing and suggest preliminary anti-senescence and anti-inflammatory effects of tomatidine in late-life interventions.

However, the conclusions drawn by the authors extend beyond the evidence provided. Specifically, the claim that tomatidine is a safe, orally bioavailable senotherapeutic with strong translational potential to extend healthspan is not sufficiently supported. The study does not include lifespan analyses, nor does it provide toxicological or pharmacokinetic assessments to establish long-term safety, tolerability, or exposure levels. In the absence of such data, the assertion of safety and healthspan extension remains speculative. At present, the manuscript should more cautiously frame tomatidine as a compound with preliminary anti-senescence and anti-inflammatory activity in aged mice, rather than as a validated senotherapeutic with proven safety and healthspan benefits.

In addition, while the authors report reductions in senescence-associated markers and neuroinflammation, these observations remain largely descriptive and correlative. The manuscript does not establish or validate a direct mechanistic link between tomatidine and neuronal function. Pathway analyses, neuron-specific functional assays, and causal validation experiments are missing, which restricts the strength of the conclusion that tomatidine can improve healthspan and cognitive function. To substantiate this claim, future studies should incorporate mechanistic experiments that directly connect tomatidine's molecular actions to neuronal physiology and cognition—for example, electrophysiological recordings, synaptic plasticity assays, neuron-targeted senescence reporters, or pathway dependency tests (e.g., Nrf2, mitophagy, NF- κ B). Such data would be essential to move beyond correlative observations and firmly establish tomatidine as a candidate therapeutic for age-related cognitive decline.

Minor Concerns

1. Tomatidine purity specification: The manuscript does not state the purity of the tomatidine compound used in the dietary supplementation experiments. This omission is critical for reproducibility and interpretation. Please specify the source and purity grade of the tomatidine used (e.g., {greater than or equal to}98% purity by HPLC).
2. QC control data: Key performance metrics of the screening model, such as Z'-factor, signal-to-background (S/B) ratio, and coefficient of variation (CV), are not reported. In addition, the scRNA-seq dataset lacks essential quality control information. Standard QC metrics such as sequencing depth per cell, number of detected genes, mitochondrial gene percentage thresholds, doublet removal criteria, and batch effect correction methods should be provided to ensure reliability and reproducibility.
3. BBB functional validation: Although the authors provide RNA-ISH, immunofluorescence, Western blot, and transcriptomic data suggesting improved endothelial health, no protein-level validation of the upregulated genes and no direct BBB permeability assays were performed. As a result, the conclusion that tomatidine preserves BBB integrity remains speculative.
4. Language and framing: In the Methods section, the authors refer to the use of "RT-PCR." Based on the description and context, it appears that the technique employed is quantitative reverse transcription PCR (qPCR or RT-qPCR), rather than conventional RT-PCR. The manuscript introduces *C. elegans* in the Discussion without providing the full name (*Caenorhabditis elegans*) at first mention. The authors should incorporate a more comprehensive review of relevant literature, critically evaluate the plausibility of their results, and more clearly acknowledge the limitations of their study.
5. Data presentation: A substantial amount of data has been relegated to the Supplementary Information, which diminishes readability and comprehension. The authors should consider moving key datasets into the main figures and reserving supplementary materials for secondary or confirmatory analyses.
6. In the method part of "human IMR90 lung fibroblasts", why did the authors use etoposide to induce senescence? Etoposide is not a specific senescence-inducer. It is usually used to induce a DNA double-strand break.
7. The description of "Senescent Ercc1^{-/-} MEFs were passed 3 times to induce senescence" is confusing. Are the Ercc1^{-/-} MEFs originally senescent or not?
8. Duplicated descriptions were shown in the "senotherapeutic screening" method part.
9. In the senolytic assay of tomatidine, why did the authors use irradiation to induce senescence? The researchers often employ various methods to induce senescence in different cell types, which should be clearly justified and explained.
10. The figure legends are not the results and should not include results interpretation; alternatively, they should include all the detailed information displayed in the figures.
11. Fig. 1c, 1e, 1g: the specific measurement parameters should be shown in the panel images.
12. Fig. 2f: The IF images should indicate where the epidermis is and where the dermis is, using arrows.
13. Fig. 2a, 2d, 2f, 2i show scale bars but without specific length labels on the images.
14. All the statistical quantification should clearly suggest the sample size (n), respectively.
15. Why the Fig. 2i, 2k only show epidermal cells but not dermal cells here?
16. Fig. 3a: The representative p16 and TAF staining images should also be shown here.
17. Fig. 4b: The representative p16 staining images should also be shown here.

We thank the editors and reviewers for their thoughtful and constructive evaluation of our manuscript. We are grateful for the time and expertise invested in providing detailed feedback, which has helped us clarify and further strengthen the presentation of our work. We have carefully considered all comments and have revised the manuscript accordingly. Below, we provide a point-by-point response outlining the changes made in direct response to each reviewer's concern. We believe that the revisions address all substantive issues and improve the clarity, rigor, and impact of the study.

Referee #1:

In this study, Costa et al screened a panel of natural compounds for their ability to reduce SA-bGal activity of peroxide-treated *Ercc1*^{-/-} MEFs and identified tomatidine as a previously unrecognized compound with senescence-reducing activity. In irradiated human fibroblasts, tomatidine reduced the expression of SASP genes, but not of p21 or p16, and did not cause significant cell death, overall revealing its senomorphic activity. 24-month-old mice, treated with 0.05% tomatidine for 3 months, showed a significantly reduced frailty index, as well as improved neuromuscular function and cognitive performance. In these mice, liver and skin tissues showed a reduction in senescence markers p16, p21, and TAF, as did neurons in mouse brain. Furthermore, tomatidine also suppressed brain endothelial cell senescence and increased the expression of tight junction genes. Although this study provides few mechanistic insights into the molecular pathways affected by tomatidine, its effects on reducing the SASP of treated cells, while also reducing cellular senescence systemically, are clearly presented and convincing. Overall, this is a very well-designed and carefully executed study, revealing clear and unambiguous results and a novel senotherapeutic compound with the potential to improve health, healthspan, and cognitive function during aging.

Minor points:Comment:

How were RNA-ish data evaluated and quantified?

How were TAF-positive cells quantified? These details should be included in the methods section.

Response:

We thank the reviewer for this suggestion. We have now added a detailed description of the RNA-ISH analysis pipeline and TAF quantification procedure to the Methods section, including image acquisition parameters, segmentation criteria, and quantification metrics.

Comment:

It would be important to mention whether any negative/side effects were observed in mice treated for three months with tomatidine.

Response:

We agree that it is important to be transparent about any potential negative side effects. Based on the parameters we evaluated, we did not observe any adverse effects of tomatidine. We have now added the following sentence to the first paragraph of the Results section:

“Importantly, throughout the 3-month intervention, tomatidine supplementation did not lead to any detectable adverse or negative side effects, as treated mice maintained normal behavior, body weight trajectory, and overall health.”

Referee #2:

This paper has nicely present the senotherapeutic effect of Tomatidine in aged mice. Below are a few points that will be helpful to further enhance the study:

Comment:

In extended figure 1D, the authors should also present the staining results of non-senescent cells as a negative control.

Response:

We thank the reviewer for this helpful suggestion. We have now added the staining results from non-senescent control cells to Extended Figure 1D.

Comment:

The author didn't specify the age of young mice in Figure 1, neither in the text or method.

Response:

We thank the reviewer for pointing this out. We have now added the age of the young control mice (4 months old) to both the main text and the Methods section.

Comment:

In the Extended data 4 and Figure 3, it would be informative to also present the quantification of microglia density in young mice in hippocampus and cortex. This information could tell us whether aging induces microgliosis.

Response:

We agree that this comparison is informative. To maintain consistent figure layout and flow, we have

added the quantification of microglia density in young mice to the Extended Data. Including it there keeps the figure coherent with the rest of the dataset while still providing the information the reviewer requested.

Comment:

It would be helpful to also quantify the neuronal density and synaptic density changes in CA1 or dentate gyrus after drug treatment in Figure 3.

Response:

We thank the reviewer for this insightful suggestion. We have now included quantification of both neuronal density and synaptic density in CA1 and dentate gyrus in Extended Data Figure 4.

Comment:

Figure 3E should also present a zoom out, lower mag images to show the microglia distribution/density throughout the hippocampus.

Response:

We have now incorporated lower magnification (20×) images into Figure 3E to provide a broader view of microglial distribution across the hippocampus.

Referee #3:

This manuscript provides a valuable contribution by presenting a relatively comprehensive screening pipeline and in vivo data, highlighting the potential of tomatidine as a nutraceutical candidate. The authors demonstrate that tomatidine supplementation in aged mice improves cognitive performance, reduces neuroinflammation, decreases senescence-associated markers in neurons and across multiple tissues, and enhances BBB integrity. These findings are intriguing and suggest preliminary anti-senescence and anti-inflammatory effects of tomatidine in late-life interventions. However, the conclusions drawn by the authors extend beyond the evidence provided. Specifically, the claim that tomatidine is a safe, orally bioavailable senotherapeutic with strong translational potential to extend healthspan is not sufficiently supported. The study does not include lifespan analyses, nor does it provide toxicological or pharmacokinetic assessments to establish long-term safety, tolerability, or exposure levels. In the absence of such data, the assertion of safety and healthspan extension remains speculative. At present, the manuscript should more cautiously frame tomatidine as a compound with preliminary anti-senescence and anti-inflammatory activity in aged mice, rather than as a validated senotherapeutic with proven safety and healthspan benefits.

In addition, while the authors report reductions in senescence-associated markers and neuroinflammation, these observations remain largely descriptive and correlative. The manuscript does not establish or validate a direct mechanistic link between tomatidine and neuronal function. Pathway analyses, neuron-specific functional assays, and causal validation experiments are missing, which restricts the strength of the conclusion that tomatidine can improve healthspan and cognitive function. To substantiate this claim, future studies should incorporate mechanistic experiments that directly connect tomatidine's molecular actions to neuronal physiology and cognition—for example, electrophysiological recordings, synaptic plasticity assays, neuron-targeted senescence reporters, or pathway dependency tests (e.g., Nrf2, mitophagy, NF- κ B). Such data would be essential to move beyond correlative observations and firmly establish tomatidine as a candidate therapeutic for age-related cognitive decline.

We appreciate the reviewer's overall positive assessment of our dataset and screening pipeline. Below we address the reviewer's concerns point by point.

Response to Major Concerns

Comment:

The reviewer raises concerns that our conclusions overstate safety and translational potential, noting that lifespan, toxicological, and pharmacokinetic studies were not performed. They further comment that mechanistic studies linking tomatidine to neuronal function are missing and suggest assays such as Nrf2 dependency, mitophagy analysis, NF- κ B modulation, or electrophysiology.

Response:

We thank the reviewer for these thoughtful comments and for recognizing the strength of our in vivo and screening data. We appreciate the opportunity to clarify the scope and intent of the present study.

1. Regarding safety, bioavailability, and healthspan claims:

Our goal in this manuscript is not to claim comprehensive toxicological validation or lifespan extension, but rather to report the effects of tomatidine in aged mice within a clearly defined 3-month intervention. We have revised the text to frame tomatidine more conservatively, as a compound with *preliminary senescence-reducing and anti-inflammatory activity* in late-life treatment, without implying full safety characterization or healthspan extension beyond the endpoints measured. We also clarified our biosafety statement to reflect the absence of adverse effects observed during the 3-month treatment period, without extrapolating beyond the data we collected.

2. Regarding mechanistic depth and neuronal function assays:

We thank the reviewer for the comment, but we want to clarify that establishing a full mechanistic pathway in neurons was not the aim of this study. Our goal was to test whether 3-month tomatidine supplementation improves healthspan and cognitive function in aged mice. We agree that mechanistic studies, including pathway dependency analyses (e.g., Nrf2, mitophagy, NF- κ B), electrophysiological recordings, or synaptic plasticity assays, would further elucidate how tomatidine influences brain aging. However, such experiments are beyond the scope of the current work. We have revised the manuscript to explicitly state that mechanistic evaluation will be an important direction for future work.

In terms of functional outcomes, we show consistent effects across independent assays, including reduced neuronal and endothelial senescence, reduced neuroinflammation, increased tight junction protein expression, and clear improvements in mobility and cognition. These results go beyond descriptive correlations and demonstrate functional benefits at the behavioral and tissue level.

Minor Concerns:

Comment:

Tomatidine purity specification: The manuscript does not state the purity of the tomatidine compound used in the dietary supplementation experiments. This omission is critical for reproducibility and interpretation. Please specify the source and purity grade of the tomatidine used (e.g., {greater than or equal to}98% purity by HPLC).

Response:

We thank the reviewer for this comment. The source and purity of tomatidine have now been added to the Methods section. The tomatidine used in the dietary intervention was purchased from Enzo Life Sciences (BML-GR335-0000; \geq 90% purity by TLC) and incorporated into chow by Research Diets Inc.

Comment:

QC control data: Key performance metrics of the screening model, such as Z'-factor, signal-to-background (S/B) ratio, and coefficient of variation (CV), are not reported. In addition, the scRNA-seq dataset lacks essential quality control information. Standard QC metrics such as sequencing depth per cell, number of detected genes, mitochondrial gene percentage thresholds, doublet removal criteria, and batch effect correction methods should be provided to ensure reliability and reproducibility.

Response:

We thank the reviewer for this suggestion. We have now added all key QC metrics to the Methods section, including:

- sequencing depth
- UMI and gene counts
- mitochondrial transcript thresholds
- doublet removal (DoubletFinder)
- and batch correction (Seurat v5 RPCA)
- number of high-quality cells retained

At the reviewer's request, we additionally calculated Z'-factor, S/B ratio, and CV metrics for p21 and p16 expression. As expected for zero-inflated single-gene scRNA-seq data, negative cells exhibit uniformly zero counts, producing infinite S/B values and negative Z'-factors. We therefore note that such metrics are not meaningful for scRNA-seq readouts, but we include them in the Methods for completeness and transparency.

Comment:

BBB functional validation: Although the authors provide RNA-ISH, immunofluorescence, Western blot, and transcriptomic data suggesting improved endothelial health, no protein-level validation of the upregulated genes and no direct BBB permeability assays were performed. As a result, the conclusion that tomatidine preserves BBB integrity remains speculative.

Response:

We thank the reviewer for pointing out this limitation. While we agree that assays such as dextran or Evans blue extravasation provide direct BBB permeability measurements, these experiments require fresh cohorts of treated mice and could not be performed retrospectively. However, we respectfully believe that our multimodal dataset, comprising single-cell RNA-seq, ZO-1 immunostaining, Western blot analysis of tight-junction proteins, and reduced p16/p21 expression in GLUT1+ endothelial cells, provides strong and convergent evidence of improved endothelial health. We have added a statement in the Discussion acknowledging that direct functional BBB assays would further strengthen the conclusions.

Comment:

Language and framing: In the Methods section, the authors refer to the use of "RT-PCR." Based on the description and context, it appears that the technique employed is quantitative reverse transcription PCR (qPCR or RT-qPCR), rather than conventional RT-PCR. The manuscript introduces *C. elegans* in the Discussion without providing the full name (*Caenorhabditis elegans*) at first mention.

The authors should incorporate a more comprehensive review of relevant literature, critically evaluate the plausibility of their results, and more clearly acknowledge the limitations of their study.

Response:

We thank the reviewer for these suggestions.

We have corrected the terminology to “RT-qPCR” throughout and spelled out *Caenorhabditis elegans* at first mention. Regarding the request for a “more comprehensive review of relevant literature” and additional discussion of limitations. We reviewed the manuscript and made sure that all central concepts, including cellular senescence, SASP biology, senotherapeutic strategies, brain aging, and tomatidine’s previously reported functions, are supported by comprehensive and up-to-date citations. In addition, we ensured that the revised Discussion now more explicitly acknowledges limitations and clarifies the scope of our mechanistic conclusions.

Comment:

Data presentation: A substantial amount of data has been relegated to the Supplementary Information, which diminishes readability and comprehension. The authors should consider moving key datasets into the main figures and reserving supplementary materials for secondary or confirmatory analyses.

Response:

We appreciate the reviewer’s perspective. We carefully considered the suggestion but believe the current structure best preserves clarity and narrative flow. All datasets needed to understand and support the main conclusions are already presented in the main figures. The supplementary figures contain additional supportive or exploratory analyses that would overcrowd the main figures without improving conceptual clarity. For these reasons, we prefer to keep the existing figure placement. We also note that in EMBO Molecular Medicine, main and Extended Data Figures appear online in sequence, ensuring that readers have easy and direct access to all data.

Comment:

In the method part of "human IMR90 lung fibroblasts", why did the authors use etoposide to induce senescence? Etoposide is not a specific senescence-inducer. It is usually used to induce a DNA double-strand break.

Response:

We thank the reviewer for this question. Etoposide is widely used in the senescence field to induce DNA damage driven cellular senescence, including in IMR90 fibroblasts (e.g.s PMID: 27048913;

21979375; 28230051; 28976970; 37821702) . Etoposide induces DNA damage by interfering with topoisomerase II and DNA double-strand breaks reliably trigger cell-cycle arrest and the senescence program. Importantly, in this study, we employed multiple well-established senescence-induction methods (irradiation, oxidative stress and etoposide), demonstrating that the senomorphic effect of tomatidine is not dependent on a single model.

Comment:

The description of "Senescent Ercc1^{-/-} MEFs were passed 3 times to induce senescence" is confusing. Are the Ercc1^{-/-} MEFs originally senescent or not?

Response:

We appreciate the opportunity to clarify. We have added the following explanation to the Methods section:

"Ercc1^{-/-} MEFs are not senescent at isolation. They are initially expanded at 3% O₂ to limit oxidative stress. To induce senescence for screening experiments, cells were transferred to 20% O₂ and serially passaged three times, leading to DNA-damage accumulation and a stable senescent phenotype."

Comment:

Duplicated descriptions were shown in the "senotherapeutic screening" method part.

Response:

We thank the reviewer for this observation. We carefully reviewed the Methods section and ensured that no redundant descriptions remain. While we did not identify duplicated text, we streamlined the section for clarity wherever possible.

Comment:

In the senolytic assay of tomatidine, why did the authors use irradiation to induce senescence?

Response:

We thank the reviewer for the question. Irradiation is one of the most widely used and well-established senescence-induction methods, particularly for senolytic studies. As noted above, we intentionally used several senescence models to ensure that tomatidine's effects were not restricted to one specific senescence-induction mechanism. Each induction method is clearly indicated.

Comment:

The figure legends are not the results and should not include results interpretation; alternatively, they should include all the detailed information displayed in the figures.

Response:

We agree and have revised the figure legends to remove interpretative statements.

Comment:

Fig. 1c, 1e, 1g: the specific measurement parameters should be shown in the panel images.

Response:

We appreciate the reviewer's concern but are unsure which additional parameters are being requested, as each graph already clearly states what was measured and how the data were quantified.

Comment:

Fig. 2f: The IF images should indicate where the epidermis is and where the dermis is, using arrows.

Response:

Done. Arrows indicating epidermis and dermis have been added.

Comment:

Fig. 2a, 2d, 2f, 2i show scale bars but without specific length labels on the images.

Response:

We thank the reviewer for this comment. While sample sizes and scale information were already provided in the figure legends, we have now also added length labels directly to all scale bars in the images as requested.

Comment:

All the statistical quantification should clearly suggest the sample size (n), respectively.

Response:

All sample sizes (n) are already provided in the figure legends. We have double-checked and ensured these are clearly indicated in every panel.

Comment:

Fig. 3a: The representative p16 and TAF staining images should also be shown here.

Response:

We now include p16 and TAF staining images in the main figure.

Comment:

Fig. 4b: The representative p16 staining images should also be shown here.

Response:

These images have been added to the Extended Data as requested.

23rd Jan 2026

Dear Dr. Jurk,

Thank you for submitting your revised study. We have now received the reports from the two referees who were asked to evaluate your revised manuscript. As you will see below, while referee #2 is overall satisfied with the revisions, referee #3 regrets the lack of deeper mechanistic insight. We have further discussed the manuscript and referee reports within the team and with the referees, and agreed that at this stage, the finding that tomatidine reduces senescence and inflammation and improves cognitive function is a sufficiently significant advance to warrant further consideration as a short report, pending revisions to address the comments on Figure 3.

Please also address the following editorial matters:

1/ Manuscript text:

- Please remove the yellow highlights and indicate in track changes mode any new modification.
- "Material and Methods" should be renamed "Methods".
 - o Animals: please indicate the origin and gender of the young mice.
 - o Cells: for all, please indicate the origin of the cells, if they were authenticated and tested for mycoplasma contamination.
 - o Statistics: please provide a statement on sample size, randomization, blinding and inclusion/exclusion criteria
- BioRender: remove from figure legends and add in a dedicated "Graphics" section in the Methods:
"Graphics:
(some of the... OR Figure #... OR synopsis) Graphics were created with BioRender.com."
 - Data availability should be placed after the Methods. Please remove "The raw data for this figure will be made available by the corresponding author upon reasonable request." Please note that the data must be public at the time of acceptance.
 - Acknowledgments: Please ensure that the funders list in our system is complete and accurate (it should match the information provided in the manuscript). Currently, U19 AG056278 is missing in our system. Should the Glenn Foundation for Medical Research, FCT scholarship 2022.11293. BD and FLAD award also be added to the list in our system?
 - Please remove author contributions from the manuscript file. You will be asked to provide CRediT (Contributor Role Taxonomy) terms in the submission system. These replace a narrative author contribution section in the manuscript.
 - References: please correct the format to alphabetical order, 10 author names listed before et al, please remove DOIs for published articles.

2/ Figures:

- Please remove the figures from the manuscript text and upload them as separate, high resolution figure files. The legends should be in the manuscript text. Please also upload the EV figures as high resolution figure files and place the legends after Tables 1 - 3, under the heading Expanded View Figure Legends. The figures should be renamed Figure EV1 etc.
- Please make sure that all figures/figure panels are referenced in the text in sequential order (currently, Table 2 is called out before Table 1, and callouts are missing for Fig 4e-k).
- Please address the queries from our data editors in the figure legends:
 1. Please define the annotated p values ****/***/**/* as well as provide the exact p-values for the same in the legend of figure 1B, C, E, G; 2B, C, E, G, H, K; 3A, B, C, D, E, G; 4C, D, F, H; EDF 1F, G, H, I, L; EDF2 A-E; EDF4 C-F; as appropriate.
 2. Please note that the white arrows are not defined in the legend of figures 2A, D, F, I; 3A-C. This needs to be rectified.

3/ Please provide a complete author checklist, which you can download from our author guidelines. Please insert information in the checklist that is also reflected in the manuscript. The completed author checklist will also be part of the RPF.

4/ Source data: please provide individual excel file for each figure panel, that should then be zipped into 1 figure file. Figures 2 and 4 have no panel j, please correct. Please also update the Source Data checklist.

5/ Every published paper includes a 'Synopsis' to further enhance discoverability. They include a short stand first (maximum of 300 characters, including space) as well as 2-5 one-sentences bullet points that summarizes the paper. Please write the bullet points to summarize the key NEW findings. They should be designed to be complementary to the abstract - i.e. not repeat the same text. We encourage inclusion of key acronyms and quantitative information (maximum of 30 words / bullet point). Please use the passive voice.

Please also provide a visual abstract to illustrate your article as a PNG file 550 px wide x 300-600 px high.

6/ Include a Reagents and Tools Table as part of the Methods section, which can be downloaded from our author guidelines.

7/ As part of the EMBO Publications transparent editorial process initiative (see our Editorial at <http://embomolmed.embopress.org/content/2/9/329>), EMBO Molecular Medicine will publish online a Review Process File (RPF)

to accompany accepted manuscripts.

This file will be published in conjunction with your paper and will include the anonymous referee reports, your point-by-point response and all pertinent correspondence relating to the manuscript. Let us know whether you agree with the publication of the RPF.

I look forward to receiving your revised manuscript.

Yours sincerely,

Lise Roth

***** Reviewer's comments *****

Referee #2 (Comments on Novelty/Model System for Author):

The revised manuscript has addressed the comments adequately. One minor comment is that the IBA1 staining in Figure 3d is not matching between treatments. The image in the Vehicle group has much less DAPI compared to the Tomatidine group, suggesting these images might not have been captured in the same hippocampal region. Furthermore, the staining in these two images is inconsistent with the quantification. It appears that the microglial density in the Tomatidine group might be similar to, or even higher than Vehicle group. Please revise this figure.

Referee #3 (Remarks for Author):

In the previous review, I recommended that the authors include mechanistic experiments that more directly link tomatidine's molecular actions to neuronal physiology and cognitive function. The authors now clarify that establishing a comprehensive mechanistic pathway in neurons was not within the intended scope of their study. Given this limitation, I remain uncertain whether the current depth of mechanistic investigation is sufficient to meet the standards expected for publication in EMM.

The authors addressed the editorial issues.

11th Feb 2026

Dear Dr. Jurk,

Thank you for submitting your revised files. Before I can proceed with acceptance, there are a few minor issues that should still be addressed:

1. Please provide a point-by-point response to the referees' comments. Please also explain why the original images were changed in Fig 3D and confirm that the associated quantification and source data remain correct.
2. Data Availability: please remove "All other data supporting the findings of this study are available within the manuscript and its Source Data files or Supplementary Information."
3. Checklist:
 - a. Please fill in the section Ethics/ Studies involving animals.
 - b. Please fill in the Data Availability section, right column.
 - c. You filled the section Data availability / computational models; please clarify and provide the relevant accession number/link.
4. Staining: please indicate in the methods and/or in the figure legends how many field were quantified per animal.
 - a. Please make sure that scale bars are provided and defined for all pictures.
5. Please indicate reuse of control in the figure legend of Fig EV1b/c.
6. Source Data:
 - a. Please check your source data for figure 4i (the last three rows are identical between old and treated animals).
 - b. Check labeling of the source data provided for Figure EV1D.
 - c. Check labeling of the source data provided for Fig EV4.

I look forward to receiving your revised manuscript.

Yours sincerely,

Lise Roth

Referee #2

The revised manuscript has addressed the comments adequately. One minor comment is that the IBA1 staining in Figure 3d is not matching between treatments. The image in the Vehicle group has much less DAPI compared to the Tomatidine group, suggesting these images might not have been captured in the same hippocampal region.

Furthermore, the staining in these two images is inconsistent with the quantification. It appears that the microglial density in the Tomatidine group might be similar to, or even higher than Vehicle group. Please revise this figure.

Response:

We thank the referee for this careful observation.

We have re-examined all original image sets. All sections were obtained from anatomically matched hippocampal regions and acquired using identical imaging settings. Different hippocampal subregions were not analyzed between groups.

We agree that the originally selected Vehicle image displayed weaker DAPI intensity, which may have given the impression of regional mismatch. We have therefore replaced the representative images in Figure 3d with new images from the same anatomical level that show comparable DAPI intensity between groups and more accurately reflect the quantified data.

Importantly, quantification was performed blinded across multiple sections per animal and remains unchanged.

Referee #3:

In the previous review, I recommended that the authors include mechanistic experiments that more directly link tomatidine's molecular actions to neuronal physiology and cognitive function. The authors now clarify that establishing a comprehensive mechanistic pathway in neurons was not within the intended scope of their study. Given this limitation, I remain uncertain whether the current depth of mechanistic investigation is sufficient to meet the standards expected for publication in EMM.

Response:

We thank the referee for raising this important point.

While a comprehensive delineation of neuron-intrinsic signaling pathways downstream of tomatidine would further extend the mechanistic framework, the primary aim of this study was to determine whether tomatidine functions as a senotherapeutic compound in aged mice and whether reduction of senescence burden translates into improved cognitive outcomes.

In this context, we demonstrate that tomatidine reduces p16- and p21-positive senescent cells, decreases endothelial senescence, improves tight junction protein expression and blood-brain barrier integrity, attenuates neuroinflammatory signatures, and significantly enhances cognitive performance in aged mice.

As noted by the editors, these findings are considered a significant advance appropriate for the Short Report format.

25th Feb 2026

Dear Dr. Jurk,

Thank you for addressing the last editorial issues. I am pleased to inform you that your manuscript is accepted for publication and is now being sent to our publisher to be included in the next available issue of EMBO Molecular Medicine.

You may qualify for financial assistance for your publication charges - either via a Springer Nature fully open access agreement or an EMBO initiative. Check your eligibility: <https://link.springer.com/journal/44321/how-to-publish-with-us>

Yours sincerely,

Lise Roth

>>> Please note that it is EMBO Molecular Medicine policy for the transcript of the editorial process (containing referee reports and your response letter) to be published as an online supplement to each paper. If you do NOT want this, you will need to inform the Editorial Office via email immediately. More information is available here: <https://link.springer.com/partners/embo-press/editorial-policies#Peer%20review>